# Buffer optimization of siRNA-lipid nanoparticles mitigates lipid oxidation and RNA-lipid adduct formation

Daniel A. Estabrook [1] ✉, Lihua Huang[2], Olivia R. Lucchese [1], Dylan J. Charland[1], Zhao Yu[2], Fareed Bhasha Sayyed[3], Jonas Y. Buser[2], Younghoon Oh [1], Xingyan Liu[1], Harmon A. Johnson[1], Kenneth G. Rodriguez [1], Noah A. Wambolt [4], Sonia A. Corba [4], Geoffrey T. Nash [2], Dennis Yang[2] & Tingting Wang[2]

Lipid nanoparticles are a versatile class of clinically approved drug delivery vehicles, particularly for nucleic acid cargoes. Despite this, these materials often suffer from instability issues that limit shelf-life or necessitate storage at ultra-cold temperatures. Herein, we demonstrate that the oxidation of unsaturated hydrocarbons within ionizable lipid tails results in the production of a dienone species that changes the conformation of the lipid tail and generates an electrophilic degradant that reacts with neighboring siRNA cargoes to produce siRNA-lipid adducts. This mechanism highlights the interplay between lipid degradation, colloidal instability, RNA-lipid adduct formation, and loss of bioactivity. In this work, we show that revised drug product matrixes, including mildly acidic, histidine-containing formulations, can improve room temperature stability of siRNA-lipid nanoparticles by mitigating these oxidative degradation mechanisms.

Nucleic acid-based therapies have become increasingly successful in recent years due to a better understanding of how to optimize nucleic acid cargo, as well as the vehicle in which it is encapsulated. A clear example of this is the rapid development of the mRNA-lipid nanoparticle (LNP) vaccines against the SARS-CoV-2 virus, i.e., the BNT162b2 (Pfizer/BioNTech) and mRNA-1273 (Moderna) vaccines. The development of Alnylam's siRNA-LNP-based therapy, Onpattro, was foundational in establishing structure-property relationships in RNA-LNP formulations. In 2018, Onpattro became the first FDA-approved RNA-LNP, and with its success came a better understanding of ionizable lipid chemistry, hepatic nanoparticle-mediated delivery, and clinical/regulatory development. However, the inherent complexity of these multicomponent RNA-LNP drug products continues to be a long-standing challenge, from production to stability. In fact, an early hurdle for these therapies was the requirement of mRNA-LNP vaccines to be distributed at (ultra)cold storage temperatures to the hospital or clinic. Shelf-life and storage challenges are reflective of colloidal stability (e.g., mitigating increases in particle size or polydispersity), cargo stability (e.g., maintaining RNA integrity), cargo leakage, and lipid stability (e.g., avoiding chemical degradation)[1]. Notably, these stability issues can be intertwined with one another—for example, Packer et al. highlighted a mechanism in which tertiary amine N-oxidation in ionizable lipids leads to lipid-mRNA adduct formation[2]. Knowledge of these mechanisms is prudent for the continued development of RNA-LNP therapies, particularly when the chemical instability of lipids results in loss of product efficacy or becomes a safety risk.

Onpattro employs a lipid cocktail of DLin-MC3-DMA (MC3, **1**, Fig. 1), PEG-2000-C-DMG, DSPC, and cholesterol. This formulation has since become a commonly used benchmark formulation in LNP studies, both in academic and industrial settings. However, the instability of the MC3 lipid is well-known, with the ester linker being susceptible to hydrolysis and the degrees of unsaturation being prone to

[1]Lilly Seaport Innovation Center, Boston, MA, USA. [2]Eli Lilly and Company, Indianapolis, IN, USA. [3]Eli Lilly Services India Pvt Ltd., Bengaluru, India. [4]Eurofins Lancaster Laboratories Professional Scientific Services, LLC, Lancaster, PA, USA. ✉e-mail: estabrook_daniel@lilly.com

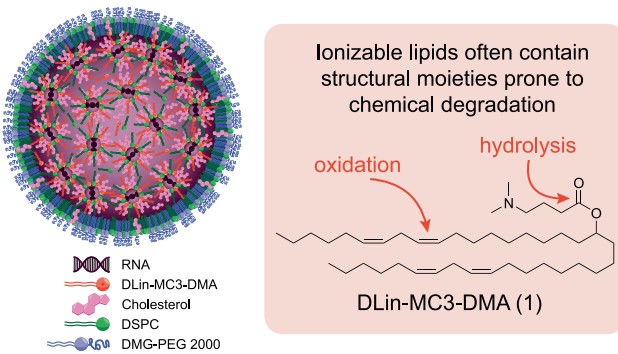

**Fig. 1 | Overview of the lipid nanoparticles employed and the drug product formulation strategies described herein.** siRNA-lipid nanoparticles are generated using a four-component lipid mixture consisting of 1,2-distearoyl-sn-glycero-3-phosphocholine (DSPC), 1,2-dimyristoyl-rac-glycero-3-methoxypolyethylene glycol-2000 (DMG-PEG-2000), cholesterol, and DLin-MC3-DMA **1**.

oxidation[3]. Like many kinetically stabilized nanoparticle systems, Onpattro can also suffer from particle instability over time in response to factors like elevated temperatures or mechanical stress. For example, while Onpattro has a shelf-life of 36 months at 2–8 °C, it falls to only 14 days when stored at room temperature[4]. During development, Alnylam researchers hypothesized that the formation of aggregations was an "inherent property of the LNP as a reaction to mechanical stress [during shipping]" that may manifest in "a white coating... at or near the liquid-headspace interface"[3]. When employing MC3 within mRNA-LNP formulations, Kamiya et al. demonstrated that environmental stresses, including freezing without a cryoprotectant, exposure to vibration, or exposure to light, all resulted in reduced protein expression, possibly due to lipid or mRNA drug substance oxidation[5]. The importance of accurately characterizing intact ionizable lipid, as well as its related impurities, has prompted analytical method development[6].

In addition to ensuring drug product safety and efficacy, a mechanistic understanding of RNA-LNP stability challenges may help to rationally inform lipid chemistries. For example, while Onpattro's MC3 lipid carries its tertiary amine at the head group, Moderna's SM-102 and Pfizer/BioNTech's ALC-0315 lipids share hydroxyl head groups connected to the amine by a two- or three-carbon linker. In the tail region, MC3 has two tails containing degrees of unsaturation, while SM-102 and ALC-0315 have three and four fully saturated tails, respectively[1,7]. While these double bonds may make MC3 more susceptible to oxidation[5], these functional motifs increase nucleic acid delivery efficiency by enhancing the fusogenicity of the lipid[8]. The kinked molecular shape of the *cis* double bond, being crucial for potency, may be a generalizable design rule, as Lam et al. recently demonstrated with a library of trialkyl ionizable lipids that stepwise replacement of *cis* double bonds with *trans* bonds results in decreased gene silencing[9]. As such, strategies to mitigate the instability of ionizable lipids may be preferred over the expectation that problematic functionalities simply will not be explored within lipid libraries.

One key parameter in assessing drug product stability is matrix optimization, including careful selection of buffer, cryoprotectants, and/or osmolytes. While siRNA-LNPs, like Onpattro, may be stored at 2–8 °C with only buffer and osmolytes, mRNA-LNP vaccines both require cryoprotectants as well, presumably due to the fragility of the

mRNA[1]. To date, phosphate and Tris have both been used as buffers across RNA-LNPs, although Tris has been preferred for frozen formulations. In fact, BNT162b2 underwent a formulation transition from a phosphate-based frozen buffer to a second-generation Tris-based buffer that allowed for longer storage at refrigerated temperatures and room temperature[10–12]. Despite broad advances in RNA-LNP formulations, it is clear there is still much to learn about how changes in the matrix can affect drug product stability.

Herein, we demonstrate that the room temperature stability of siRNA-LNPs formulated with unsaturated ionizable lipids can be improved by inclusion of mildly acidic, antioxidant-containing buffers. We show that while phosphate-based formulations limit the room temperature stability of siRNA-LNPs to two weeks, the use of a histidine-containing buffer allows for room temperature stability of 6 months to date. The stability of these nanomaterials is shown to be correlated with the oxidation of the unsaturated lipid tail, which results in the production of a dienone species. We show that these oxidative byproducts have three deleterious effects, namely, they (i) introduce conformational changes in the lipid tail, (ii) produce a hydrogen bond acceptor that non-covalently interacts with siRNA, and (iii) generate an electrophilic species that reacts with nucleophilic residues in siRNA cargo to produce siRNA-lipid adducts. This mechanism underwrites the significance of how lipid degradation can result in colloidal instability, siRNA cargo degradation, siRNA-lipid adduct formation, and overall loss of drug product potency. More broadly, we demonstrate that a thorough understanding of lipid degradation mechanisms is needed to enable systematic exploration of drug product matrixes. As nanoparticle drug products continue to advance in the clinic, the desire to screen lipid and RNA chemistries should be matched with a motivation to develop optimal formulations. These matrixes play a critical, sometimes underestimated role in dictating the shelf-life stability of multicomponent nanoparticles—a major hurdle in RNA-LNP drug product development.

## Results and Discussion
### Formulation of lipid nanoparticles
Lipid nanoparticles were formulated following the Onpattro formulation and using a model oligonucleotide, siHPRT, as a cargo. Details regarding LNP formulation, mixing, and downstream processing are provided in the materials and methods. Briefly, a four-component lipid mixture was solubilized in ethanol to a total lipid concentration of 12.5 mM (MC3: DMG-PEG-2000: DSPC: cholesterol at a molar ratio of 50: 1.5: 10: 38.5). siHPRT was solubilized in 50 mM sodium citrate, pH 5, at an N:P ratio of 6. The ethanol and aqueous solutions were then mixed via a T-junction mixer at a flow rate ratio of 3:1 (aqueous:ethanol, vol%) (Supplementary Fig. 1). After further dilution and a brief hold, siRNA-LNP solutions were then concentrated to an siRNA concentration of ~0.4–0.5 mg/mL and buffer exchanged into the appropriate storage buffer. Buffer exchange was performed primarily through tangential flow filtration (TFF), but was completed using desalting columns at small scales in some instances. After buffer exchange and prior to long-term storage, all siRNA-LNP particle sizes were 70 ± 5 nm with polydispersity indexes ≤0.2 and encapsulation efficiencies >85%.

### Colloidal stability of LNPs within phosphate buffer
To explore the impact of drug product matrix on the siRNA-LNP formulations, we initially studied 1X PBS, pH 7.4 (without calcium or magnesium). This formulation was selected because siRNA-LNPs can be kept refrigerated for long-term stability, and it is expected to closely mirror the formulation of Onpattro[3]. At the initial time point, the transparency of siRNA-LNP solutions was dependent on particle concentration, but generally was a homogenous off-white solution (Fig. 2A). Particle size was corroborated by cryoEM analysis, which confirmed a spherical morphology with a uniform, electron-dense core with few phase-separated blebs (Fig. 2B)[13–15]. Vials were then boxed to

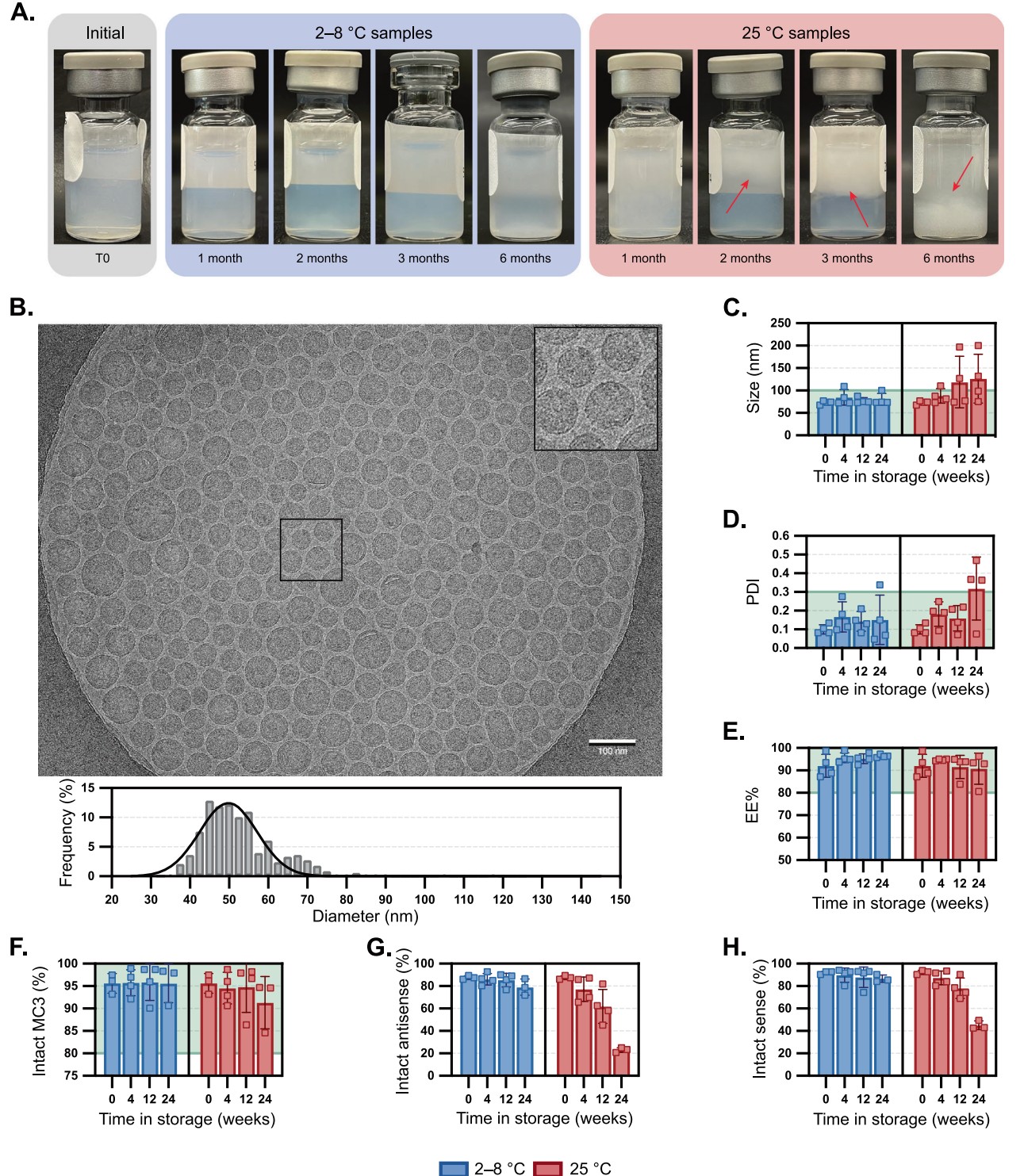

**Fig. 2 | Stability of siRNA-LNPs stored in PBS over time. A** Compilation of LNP samples photographed upon removal from storage at 2–8 °C or 25 °C for up to six months. Red arrows denote where phase separation is occurring. **B** Visualization of an LNP sample stored at room temperature for four weeks then visualized by cryogenic electron microscopy (scale bar = 100 nm). A portion of the image is magnified in the inset. LNP diameters were individually measured ($n = 326$) using image processing software, and a histogram (bin width = 2.5 nm) of relative frequency was constructed alongside a curve resulting from nonlinear regression on the data. The impact of LNP storage in PBS at 2–8 °C or 25 °C was compared to defined thresholds for the following critical quality attributes: **C** Z-average particle diameter and **D** polydispersity index by dynamic light scattering, **E** encapsulation efficiency by RiboGreen assay, and amounts of **F** intact MC3 **1**, **G** intact siHPRT antisense strand, and **H** intact siHPRT sense strand by LC/MS. Bar height and error bars represent the mean and standard deviation of four biological replicates, all of which are independently plotted as closed squares. Encapsulation efficiencies over time at 5 and 25 °C (**E**) were evaluated by linear regression analysis and had two-tailed ANOVA p-values of 0.12 and 0.45, respectively. The portion of a graph highlighted in green represents the values that meet the threshold set for the critical quality attribute. Source data are provided as a Source Data file.

minimize exposure to light and stored either at refrigerated conditions (2–8 °C) or at room temperature (RT, ~22–25 °C) over a period of up to six months. After a minimum storage time of one month, visual inspection of vials stored at RT showed early signs of particle aggregation and by two and three months, phase separation was apparent (Fig. 2A, with phase separation denoted by red arrows). We visually observed that aggregation began as small, dispersed particulates (1 month), accumulated at the air-water interface as a white film (2 and 3 months), then largely sedimented to the bottom of the vial as a white, wispy aggregate (6 months). By comparison, vials at 2–8 °C remained homogenous. In addition to macroscopic growth, dynamic light scattering analysis (DLS) showed that particle hydrodynamic diameter increased from ~73 to 127 nm and polydispersity index (PDI) increased to ≥0.30 (Fig. 2C, D). Intensity-based scattering exhibited multimodal particle size distributions (Supplementary Fig. 2), with populations ranging from 70–200 nm as well as >5 μm. To capture count rates of sub-visible particles, Micro-Flow Imaging (MFI) was performed. Particle aggregation resulted in (i) higher particle/mL count rates (1–100 μm) and (ii) a shift of macroscopic particle distributions to larger diameters (Supplementary Fig. 3). Nanoparticle tracking analysis of siRNA-LNPs held at either 5 or 40 °C over one week demonstrated a statistically significant decrease in sub-micron particles, particularly those around ~70 nm, for samples at elevated temperatures (Supplementary Fig. 4). Collectively, these data corroborate that smaller particles gradually aggregate at higher temperatures to form larger, micron-sized aggregates in phosphate-based buffers. By comparison, when stored refrigerated, mean particle sizes remained below 100 nm and mean PDIs remained below 0.2.

## Stability of ionizable lipid and encapsulated siRNA cargo

Next, we investigated the ability of siRNA to leach from the LNP vehicles over time. RiboGreen assays were performed and revealed that encapsulation efficiencies (EE%) were similar over time regardless of storage temperature (Figs. 2E, 5 and 25 °C had p-values of 0.12 and 0.45, respectively). By the same analysis, the siRNA content remained similar in all samples over time (Supplementary Fig. 5, 5 and 25 °C had p-values of 0.38 and 0.85, respectively). In addition to colloidal stability, we sought to investigate the integrity of both the ionizable lipid as well as the siRNA cargo through liquid chromatography mass spectrometry (LC/MS) analysis. We first evaluated the stability of MC3 lipid itself within 1X PBS at 5, 25 and 40 °C for upwards of four weeks (Supplementary Fig. 6). Surprisingly, rapid degradation of the lipid was observed, with >25% degradation occurring within four weeks at 5 °C, or at one week for both 25 °C and 40 °C (Supplementary Fig. 6A). The major degradation pathway was oxidation, with oxidative byproducts representing 25–30% of the degraded lipid (Supplementary Fig. 6A–C). Hydrolysis byproducts were observed but in lower amounts, but negligible in most instances (Supplementary Fig. 6A–C).

We then analyzed MC3 integrity when the lipid was formulated within siRNA-LNPs. During storage over six months, MC3 lipid integrity dropped by ~5% and ~9% at 2–8 °C or RT, respectively (Fig. 2F). Byproducts were identified and are noted within Supplementary Table 1 and Supplementary Fig. 19, which revealed that most impurities come from tail oxidation. Other byproducts, like those associated with ester hydrolysis or head group oxidation, were observed but in comparatively lower amounts. Surprisingly, many of these oxidized impurities are present even at the initial time point, indicating likely degradation of the lipid prior to particle fabrication. Overall, while the degradation of MC3 appears to be slower within nanoparticle systems than when solubilized within ethanolic PBS, it is still not avoided. Whether this slowed degradation is a result of lipids being packed within a nanoparticle or due to downstream removal of residual impurities from the LNP material (e.g., TFF) is a focus of ongoing investigation. These results corroborate the precautions necessary for handling sensitive lipids like MC3 at the manufacturing scale[3], though it is worth highlighting that many researchers may not go to these same lengths at the discovery stage.

After confirming the oxidation of MC3 within RNA-LNPs, we were curious to examine the integrity of the encapsulated siRNA payload. Previous reports have noted how oxidative lipid impurities may concurrently cause the oxidation of encapsulated mRNA[1]. Analysis of both the sense and antisense strand of the siRNA cargo revealed considerable degradation over time (Fig. 2G, H), with the level of intact sense and antisense strands falling to only 45% and 23%, respectively. Most of the degradation observed was a result of one or multiple conversions of phosphorothioates to phosphodiesters (PS-to-PO) (Supplementary Table 2). PS modifications have been identified in RNA and DNA[16], and are commonly introduced into synthetic siRNA, ASO, and microRNA therapeutics to increase serum stability, cellular permeability, and plasma protein binding[17–19], although PS modifications are less commonly explored for mRNA[20,21]. Here, we observe that PS motifs can be oxidized back to PO linkages[22]. Granted that these siRNA-LNPs have a high encapsulation efficiency and there is precedent suggesting ionizable lipids and analogous RNA cargoes tend to organize within the LNP core[23–25], we speculate that the oxidation of these two molecules also occurs within the core (as opposed to within bulk solution). These results also highlight limitations of using the RiboGreen assay that other researchers have pointed out, namely that it is not an appropriate method of detecting RNA degradation or functionality[26]. Collectively, these data demonstrate the colloidal and chemical stability issues of siRNA-LNPs, including increases in particle size and polydispersity, microparticle aggregation, and oxidation of lipids and encapsulated siRNA.

## Improving siRNA-LNP stability through drug product matrix optimization

We reflected on the stability issues of pharmaceutical excipients like polysorbate 80 that are known to undergo oxidation at the fatty acid chain[27,28]. In previous work, we studied the mechanism of polysorbate 80 oxidation and how buffer agents can impact degradation. In particular, we employed histidine-containing buffers that can mitigate excipient oxidation compared to phosphate buffers under light exposure and elevated temperatures[29]. Given the analogous oxidation problems observed with MC3, we evaluated the stability of siRNA-LNP systems within a 10 mM histidine buffer at pH 6.0. To match the osmolarity of 1X PBS, 140 mM sodium chloride was also included. At the initial time point, siRNA-LNPs in histidine-containing buffers were visually comparable to those formulated in PBS (Fig. 3A), had similar sizes (Fig. 3B, C, p-value of 0.96) and more narrow polydispersities (Fig. 3D, p-value of 0.02), as corroborated by cryoEM analysis. Interestingly, we noted that the nanoparticle morphology of siRNA-LNPs in histidine buffer had more blebs (Fig. 3B) than those in PBS, presumably due to variations in formulation pH. Compared to the visible aggregation within siRNA-LNPs in PBS, those formulated in histidine were consistently translucent, even after six months at RT (Fig. 3A). Over six months in 2–8 °C storage, there were no pronounced changes in size, PDI, or EE% (Fig. 3C–E). More strikingly, the siRNA-LNPs have remarkable colloidal stability even at RT storage, increasing by only ~11 nm over six months (73 to 84 nm, intensity data in Supplementary Fig. 7) with no notable changes in PDI (0.05 to 0.04), siRNA content (0.37 to 0.42 mg/mL) or EE% (97% to 98%). The amount of intact MC3 lipid decreased slightly over the six months with negligible differences between RT and 2–8 °C storage (94.0% and 92.0%, respectively).

Analysis of degradants suggested that while oxidation was still the primary driver of MC3 degradation, histidine largely mitigated it. Unfortunately, histidine was unable to entirely protect the siRNA cargo, which still experienced PS-PO conversion (Fig. 3G, H). These findings corroborate prior work employing antioxidants in dermatological formulations to mitigate desulfurization of PS-containing oligonucleotides[30]. Although significantly improved when compared

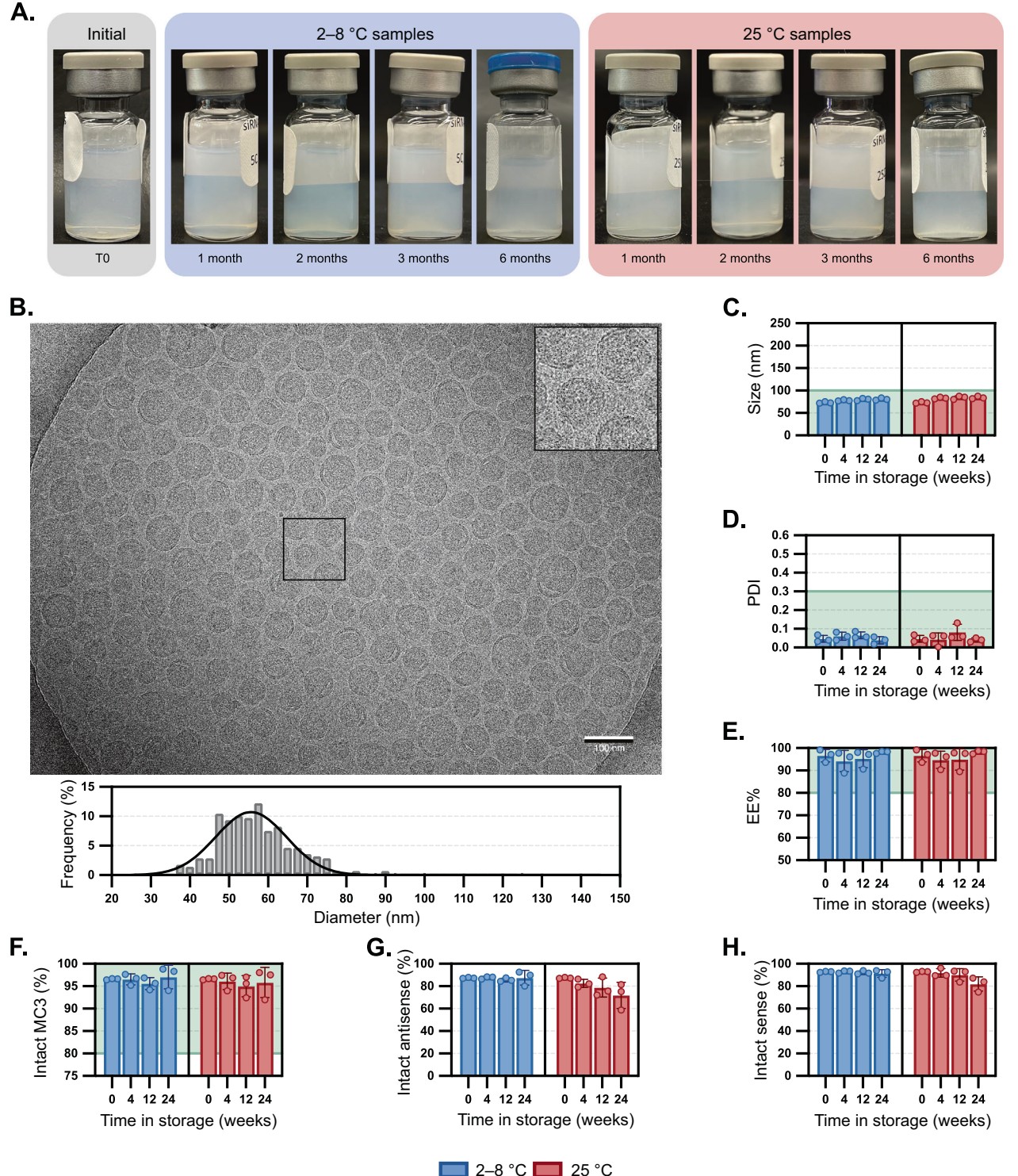

**Fig. 3 | Stability of siRNA-LNPs stored in 10 mM histidine and 140 mM sodium chloride over time. A** Compilation of LNP samples photographed upon removal from storage at 2–8 °C or 25 °C for up to six months. **B** Visualization of a LNP sample stored at room temperature for four weeks then visualized by cryogenic electron microscopy (scale bar = 100 nm). A portion of the image is magnified in the inset. LNP diameters were individually measured (n = 277) using image processing software, and a histogram (bin width = 2.5 nm) of relative frequency was constructed alongside a curve resulting from nonlinear regression on the data. The mean diameter and standard deviation are listed. The impact of LNP storage in histidine at 2–8 °C or 25 °C was compared to defined thresholds for the following critical quality attributes: **C** Z-average particle diameter and **D** polydispersity index by dynamic light scattering, **E** encapsulation efficiency by RiboGreen assay, and amounts of **F** intact MC3 **1**, **G** intact siHPRT antisense strand, and **H** intact siHPRT sense strand by LC/MS. Bar height and error bars represent the mean and standard deviation of three biological replicates, all of which are independently plotted as closed squares. Particle sizes (**C**) and PDIs (**D**) at the initial time points were compared to those stored in PBS (Fig. 2C, D) via a two-tailed Student's *t*-test assuming an unequal sample variance and had p-values of 0.96 and 0.02, respectively. The portion of a graph highlighted in green represents the values that meet the threshold set for the critical quality attribute. Source data are provided as a Source Data file.

to PBS formulation, there is room for improvement: for example, by screening other excipients, or increasing excipient concentrations (e.g., >20 mM). Similar stability profiles between MC3 and siRNA cargo suggest that the degradation mechanisms are correlated (Supplementary Fig. 8), supporting the notion that degradation of multicomponent nanoparticle formulations is a complicated and oftentimes intertwined process.

To demonstrate the generalizability of the histidine buffer platform, we systematically explored three additional cationic/ionizable lipids: DOTAP, DODMA, and DLin-KC2-DMA. We formulated siHPRT-LNPs with each lipid and buffer-exchanged the resulting particles into either PBS or the histidine-NaCl buffer. To accelerate degradation, select samples were spiked with 1 ppm of either hydrogen peroxide or metals (nickel, iron, and copper). Particles were then stored at either 25 or 40 °C for up to four weeks, and analyzed for changes in colloidal stability, siRNA payload integrity, and ionizable lipid integrity. Starting with DOTAP, we observed changes in siRNA-LNP stability for particles stored in PBS, including particle size, payload, and ionizable lipid integrity (Supplementary Fig. 9). In all instances, degradation could be mitigated by using histidine buffer. Notably, DOTAP lipid was more sensitive to hydrolysis than oxidation, which we hypothesized was due to there being a single alkene in each lipid tail and two ester linkages (Supplementary Table 3). To confirm this, we next leveraged a DODMA ionizable lipid that contains a single alkene in each lipid tail but lacks esters (Supplementary Fig. 10). Accordingly, DODMA-LNPs proved more resilient to colloidal instability and ionizable lipid degradation (Supplementary Table 4); however, payload integrity continued to prove problematic for particles stored in PBS spiked with either metals or peroxides, whereas those stored in histidine retained payload integrity across all conditions. Finally, DLin-KC2-DMA, a lipid which preceded the optimization of MC3, was employed due to its lack of esters (instead containing a cyclic acetal linker) and structurally similar bis-allylic dilinoleyl tail (Supplementary Fig. 11). DLin-KC2-DMA LNPs demonstrated dramatic colloidal and chemical degradation when stored in PBS, whereas histidine buffers again mitigated these changes over time (Supplementary Table 5); however, histidine buffers were unable to rescue samples spiked with metals. The increased sensitivity of MC3 and DLin-KC2-DMA to oxidation in comparison to DOTAP and DODMA is supported by the fact that lipids with higher degrees of unsaturation oxidize more rapidly due to the weakness of a bisallylic C−H bond compared to an allylic C−H bond[31]. For example, linoleic acid is known to oxidize faster than oleic acid[32]. However, while each cationic/ionizable lipid may differ in its particular degradation mechanisms, the ability of histidine buffers to improve product stability is shown to be generalizable across this class of LNPs. To our knowledge, this is the first demonstration of histidine-containing formulations showing such a dramatic improvement on siRNA-LNP room temperature stability.

## Oxidative degradants of unsaturated ionizable lipid tails may react with siRNA cargo

While mass spectrometry was useful in identifying the molecular weight of degradants, we moved to NMR for further structure identification via a combination of $^1$H, HSQC, COSY, and HMBC experiments. The elucidation of MC3's oxidative impurities has been the focus of previous reports and innovative methods, which have suggested oxidation of the lipid tail via epoxide formation[6]. We sourced MC3 from three vendors and analyzed the raw material by two-dimensional $^1$H-NMR. For all lots, two primary degradants were identified, both oxidative impurities in the lipid tail corresponding to either dienone (major impurity, 2, Fig. 4A) or dienol species (minor impurity, S1, Supplementary Fig. 12). These impurities include those previously identified for oxidation of linoleic acid, though the ketone is less commonly reported (note: here the ketone is drawn internal (towards head), though it could also be terminal (towards tail))[33,34]. Density

functional theory (DFT) calculations were employed to corroborate the oxidative instability of the bis-allylic C-H bond in the MC3 1 tail and to help explain the dienone 2 and dienol S1 oxidative byproducts, with the mechanism supplied in Supplementary Fig. 13. To our knowledge, this is the first time that byproducts 2 and S1 have been identified for MC3, despite its widespread use as an ionizable lipid and known oxidation challenges.

We anticipated that the oxidation of MC3 1 into E,Z-conjugated dienone 2 could be deleterious for siRNA-LNP stability in at least three ways. First, an alkene conformational change (from Z,Z to E,Z) could impact drug product potency, as previously highlighted[9], and influence lipid packing. The latter hypothesis was corroborated by correlating LNP particle size versus the amount of dienone 2 (Supplementary Fig. 14), though a more detailed investigation would be needed to understand if there is truly a causative effect. Secondly, the introduction of a hydrogen bond acceptor into the lipid tail— namely the ketone—could affect RNA-lipid interactions. The importance of hydrogen bonding interactions in ionizable lipid chemistry has previously been highlighted[35,36]. Molecular dynamics (MD) simulations and subsequent calculation of radial distribution functions between siHPRT and dienone 2 demonstrated that hydrogen-bonding between the ketone in the lipid tail and RNA is one-sixth that of the interaction between 2's tertiary amine head group and RNA (Supplementary Fig. 15A, B). This data suggests oxidative degradants may introduce new non-covalent interactions between ionizable lipids and the RNA cargo. It is worth noting that the siRNA sequence used here is entirely methylated at the 2′-OH position, but that additional MD simulations with unmodified RNA sequences demonstrated hydrogen-bonding between the RNA's native 2′OH position and the ketone of 2 (Supplementary Fig. 15C).

Finally, we hypothesized that the dienone motif within 2 could act as a strong electrophile, making the lipid byproduct prone to nucleophilic attack (Fig. 4A). We formulated a single batch of siHPRT-LNPs that was then buffer exchanged via desalting columns into three different formulations: (i) PBS (1X), (ii) Histidine (10 mM), and (iii) Tris (10 mM). LNPs were then stored at elevated temperatures of either 25 or 40 °C and analyzed over four weeks. The amount of intact MC3 1 was reduced to ~65–70% for LNPs in both the phosphate and Tris-containing buffers, while LNPs in histidine maintained >97% intact MC3 (Fig. 4B). LC/MS analysis revealed the primary degradant byproduct was E,Z-dienone 2, which was produced at a rate of ~8% over 28 days in the phosphate and Tris buffers but mitigated to <1% in the histidine buffer (Fig. 4C). A full table of MC3 and its related byproducts is supplied in Supplementary Table 6. Interestingly, LC/MS analysis of the siHPRT cargo revealed that both the sense and antisense strands were significantly lipidated—after four weeks at 40 °C, about 13% of the antisense strand was lipidated after encapsulation in LNPs within phosphate buffer, and about 68% of the antisense strand was lipidated in LNPs within Tris buffer (Fig. 4E). By comparison, only ~0.1% of lipidated antisense strand was detected for LNPs in histidine buffer (Fig. 4E). In addition, unwanted PS-PO conversion was noted, occurring one, two or even three times per strand, decaying the amount of intact antisense strand to 12% or 1% when LNPs were stored in phosphate or Tris-containing buffers, respectively (Fig. 4F). Histidine buffers retained 85% intact antisense strand over the same stressed conditions, a marked improvement (Fig. 4F). A full table of siHPRT sense, antisense strands and related byproducts, including siRNA-lipid adduct formation, is supplied in Supplementary Table 7. While lipid oxidation, dienone 2 generation, and siRNA lipidation are all correlated (Fig. 4A–C), we sought further evidence that dienone 2 was responsible for reactivity with encapsulated cargo. It is known that Michael acceptors can react with positions in nucleobases[37,38], as well as phosphorothioate linkages in the backbone[39]. Tandem MS on MC3-modified sense and antisense strands first revealed a distribution of molecular weights for the two strands that differed by ~15.98 daltons,

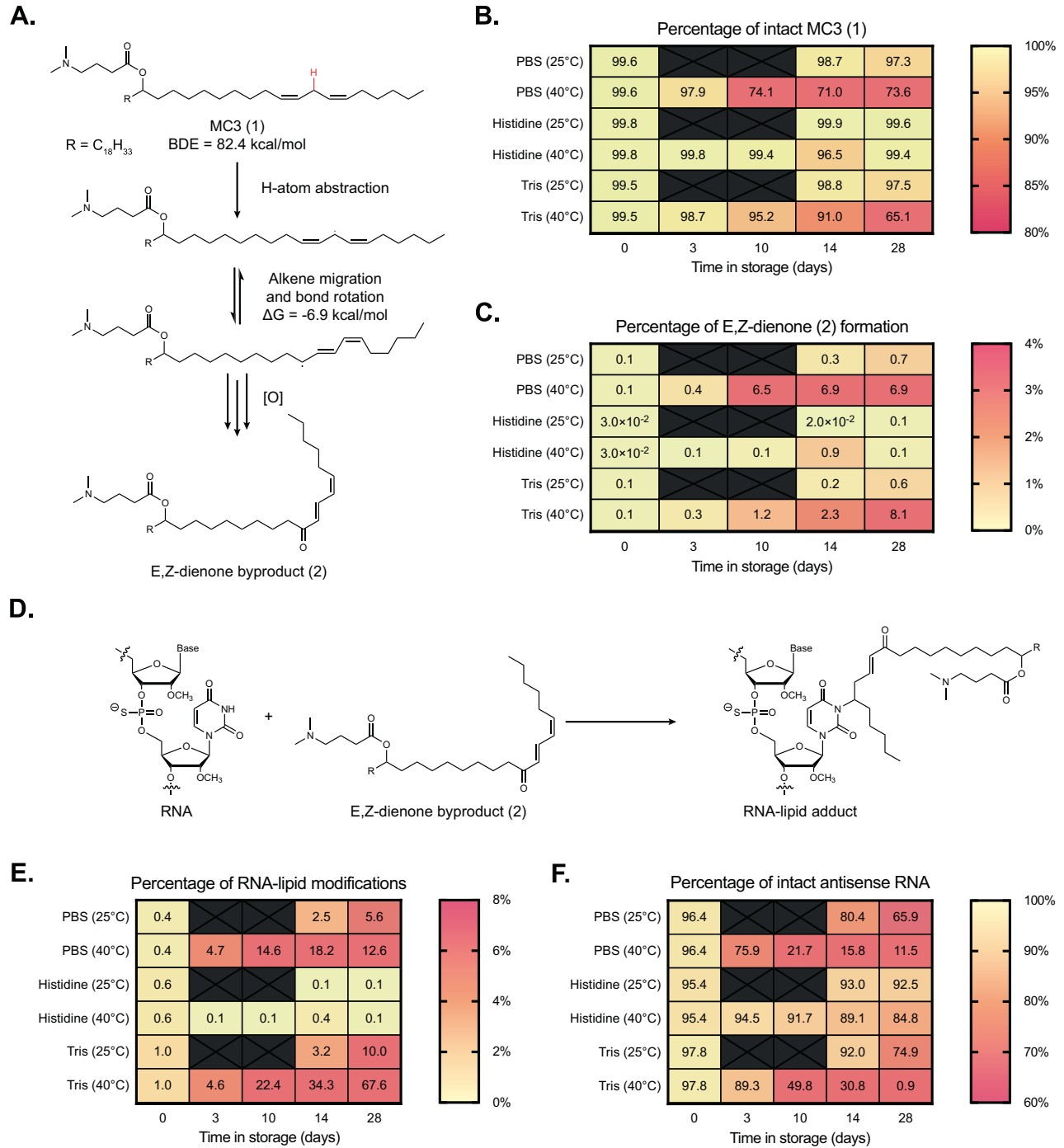

**Fig. 4 | The effect of ionizable lipid oxidation on siRNA-LNP stability and siRNA-lipid adduct formation. A** Proposed mechanism of MC3 **1** oxidation to form E,Z-dienone byproduct **2**. Full mechanism supplied in Supplementary Fig. 10. siHPRT-containing LNPs formulated in PBS, histidine, and Tris were compared to assess buffer impact on **B** MC3 **1** integrity and **C** byproduct **2** formation. **D** The electrophilic nature of **2** is observed to react with nucleic acid cargoes, generating RNA-lipid adducts. Additional testing quantified the percentages of **E** RNA-lipid adducts and **F** intact siHPRT antisense strand. Source data are provided as a Source Data file.

indicative of the PS-PO conversions discussed previously (Supplementary Fig. 16A, B). More strikingly, MS analysis of lipidated siRNA byproduct revealed molecular weight distributions that were larger than the parent molecular weights by 655.57 daltons—a value within instrumental error for *E,Z*-dienone **2**'s exact mass (655.59 ± 0.06 daltons) (Supplementary Fig. 16A, B).

Lipidation of siRNA occurred up to two times per strand, albeit less commonly (e.g., 13.2% and 2.4% of antisense strand was lipidated once or twice, respectively, after 6 months of storage in PBS 1X

(Supplementary Fig. 16, Supplementary Table 2)). Although the exact site of lipidation is still under investigation, we believe that modification through nucleobase addition is most likely, as (i) addition via phosphorothioate would result in a 1 dalton-larger molecular weight shift, and (ii) one would expect that the siRNA strand's molecular weight distributions associated with PS-PO conversion would be less likely to form lipid adducts, which was not observed (Supplementary Fig. 16A, B). Collectively, these data support a mechanism of unintentional RNA lipidation: oxidation of the unsaturated lipid tails within

**Table 1 | Performance of siRNA-LNP formulation buffers assessed by critical quality attribute (CQA) thresholds**

| | | Critical Quality Attribute | | | | | | |
|---|---|---|---|---|---|---|---|---|
| | | Passes Visual Inspection (Yes/No) | Size (nm) | PDI | Encapsulation Efficiency (%) | Intact MC3 (%) | Intact Antisense Strand (%) | Intact Sense Strand (%) |
| siHPRT-LNP Formulations | 1X PBS pH 7.4 | N | 287 | 0.44 | 53 | 70 | 45 | 66 |
| | 1X PBS + 50 µM EDTA pH 7.4 | Y | 64 | 0.09 | 96 | 99 | 91 | 97 |
| | 1X PBS + 10 mM Imidazole pH 7.4 | Y | 75 | 0.19 | 93 | 98 | 66 | 90 |
| | 1X PBS + 10 mM Methionine pH 7.4 | Y | 63 | 0.07 | 97 | 99 | 91 | 97 |
| | 1X PBS + 10 mM Na-Ascorbate pH 7.4 | N | 164 | 0.30 | 96 | 99 | 82 | 92 |
| | 1X PBS + 10 mM Tryptophan pH 7.4 | Y | 67 | 0.14 | 97 | 99 | 82 | 97 |
| | 1X PBS + 1 mM NAc-Tryptophan pH 7.4 | N | 141 | 0.16 | 82 | 84 | 26 | 54 |
| | 10 mM Histidine + 140 mM NaCl pH 6.0 | Y | 71 | 0.08 | 99 | 95 | 91 | 97 |
| | 10 mM Histidine + 280 mM Sucrose pH 6.0 | Y | 80 | 0.27 | 97 | 96 | 87 | 94 |
| | 10 mM Acetate pH 5.0 | Y | 67 | 0.18 | 93 | 71 | 5 | 21 |
| | 10 mM Citrate pH 7.0 | Y | 83 | 0.18 | 94 | 96 | 88 | 96 |
| | 10 mM Tris pH 6.0 | Y | 86 | 0.16 | 99 | 97 | 75 | 92 |

All samples were stored at room temperature for four weeks. Samples pass visual inspection if there are no observed heterogeneity or particulate formation observed.

MC3 **1** results in the production of electrophilic *E,Z*-dienone byproduct **2**, which then reacts with neighboring RNA to form RNA-lipid adducts (Fig. 4D). DFT calculations were performed to understand the formation of RNA-lipid adducts using a model base guanine, with a lipid model system (3E,5Z)-hepta-3,5-dien-2-one. The activation enthalpies were determined as 20.0 and 24.7 kcal/mol for 1,6 and 1,4-addition, respectively, indicating that 1,6 addition is favorable.

This mechanism is conceptually similar to that reported recently[2], with a number of key distinctions: in the cited work (i) the ionizable lipids were primarily saturated, (ii) the oxidative byproducts highlighted were produced through head group N-oxidation, (iii) the problematic electrophilic degradants were aldehydes, (iv) lipid adducts were formed with mRNA, a single-stranded nucleic acid with solvent-exposed nucleobases, and (v) formulation strategies to mitigate this degradation were not proposed. With knowledge of the proposed mechanism and the implications of shutting down lipid oxidation, we envisioned that many other excipients could be used to improve drug product stability. We formulated a large batch of siHPRT-LNPs and buffer-exchanged aliquots through small-scale desalting columns into a number of formulations (Table 1), with excipients including EDTA, imidazole, methionine, sodium ascorbate, tryptophan, NAc-tryptophan, acetate, citrate, and Tris. After an abbreviated study of four weeks at room temperature, the critical quality attributes of each siHPRT-LNP solution were measured and summarized in Table 1. Aside from histidine, the excipients that promoted stability best included methionine, tryptophan, EDTA, and citrate. We believe that the latter two excipients impart particle stability by chelating trace metals that are otherwise capable of catalyzing lipid oxidation, as forced

degradation experiments on MC3 **1** in solution (not formulated in an LNP) demonstrated susceptibility to oxidation in solutions containing 1 ppm nickel, iron and copper (Supplementary Fig. 17A, C).

## Optimized siRNA-LNP formulations retain potency after extended storage at room temperature

Finally, we tested the knockdown efficiency of siHPRT-LNPs within HeLa cells after LNPs were stored for one month at either 2–8 °C or at room temperature. A head-to-head comparison was again done between two buffers, 1X PBS (pH 7.4) and 10 mM histidine and 140 mM NaCl (pH 6.0). siHPRT samples incubated with or without the transfection agent RNAiMax were used as positive and negative controls, respectively. On the day of transfection, fresh LNPs were generated, buffer exchanged into 1X PBS, and compared to aged LNP samples. After a 24-hour transfection, RNA was isolated via a Cells-to-CT assay kit and *HPRT* gene expression was quantified by qPCR (Fig. 5A). We observed that LNPs stored at refrigerated temperatures in either PBS or histidine-containing buffers had similar knockdown efficiencies to that of fresh LNPs. However, LNPs stored in PBS at room temperature lost potency, with no detectable IC50 at these concentrations. On the other hand, LNPs stored in histidine buffer at room temperature for one month (red line) remarkably retained potency, with IC50 values similar to those of fresh or refrigerated LNPs (Fig. 5B). For example, at a concentration of 0.1 nM siRNA, *HPRT* expression is effectively knocked down to 42 ± 2% by fresh LNPs, as well as LNPs refrigerated in PBS (ns, p-value of 0.65), and LNPs refrigerated or at room temperature in histidine buffer (ns, p-values of 0.85 and 0.79). Conversely, *HPRT* expression after incubation with LNPs at room temperature in PBS has

**A.**

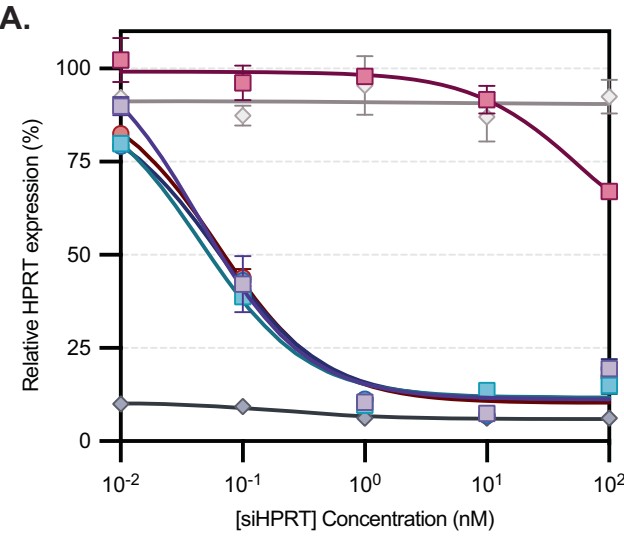

**B.**

| Sample names and symbols | IC50 (nM) |
|---|---|
| Free siHPRT | N/A |
| RNAiMax transfected siHPRT | 0.219 |
| Fresh (1X PBS) | 0.045 |
| 4W @ 2–8 °C (1X PBS) | 0.045 |
| 4W @ 25 °C (1X PBS) | N/A |
| 4W @ 2–8 °C (10 mM His, 140 mM NaCl) | 0.063 |
| 4W @ 25 °C (10 mM His, 140 mM NaCl) | 0.063 |

**Fig. 5 | Storage temperatures and buffer choice impact siRNA-LNP knockdown.**
**A** Relative mRNA expression was measured in response to increasing sample concentration ($n = 2$ independent replicates), normalized to an untreated control. Except for fresh LNP formulated on the day of assay, all LNP samples were stored at 2-8 °C or 25 °C for four weeks in 1X PBS or 10 mM histidine buffer. siHPRT incubated with RNAiMax and free siHPRT were used as positive and negative controls, respectively. Markers and error bars represent the mean and standard deviation between experimental duplicates. At 0.1 nM siRNA, *HPRT* expression was compared across groups to fresh LNPs using a two-tailed Student's *t*-test assuming unequal sample variance. LNPs stored in PBS at refrigerated temperature, or LNPs stored in histidine buffer at either refrigerated or room temperature, showed no significant difference (*p*-values > 0.05), while LNPs at room temperature in PBS resulted in negligible knockdown ($p = 0.02$). **B** IC50 values for the samples were determined using nonlinear regression. N/A refers to samples for which a plateau was not established over the tested concentrations. Source data are provided as a Source Data file.

negligible knockdown, with a relative *HPRT* expression of 96% (p-value of 0.02 when compared to fresh LNPs). These data support that siRNA-LNPs stored in histidine-containing buffers retain their biological function even at elevated temperatures, while those stored in phosphate-containing buffers undergo a temperature-dependent loss in potency.

In summary, we have shown that degrees of unsaturation in ionizable lipids are prone to degradation through an oxidative pathway that results in electrophilic degradants. These problematic impurities can react with neighboring RNA cargoes, forming RNA-lipid adducts. Degradation is observed at high levels (>40 mol%) for LNPs formulated in common phosphate-based matrixes. We show that this degradation can be mitigated by optimizing the drug product matrix; for example, through the use of mildly acidic, antioxidant-containing histidine formulations. Histidine-based formulations are shown to improve siRNA-LNP stability, ultimately extending the room temperature shelf-life from two weeks to upwards of six months to date. Finally, it is shown that the bioactivity of model siRNA-LNPs can be retained for at least a month at room temperature, illustrating the significance of ensuring adequate colloidal, lipid, and siRNA drug substance stability in siRNA-LNP systems.

This work highlights challenges that academic and industrial researchers must be cognizant of when developing these multi-component drug products, and the ability of drug product matrix optimization to solve them. However, a limitation of the work described herein is that it primarily focuses on siRNA-LNPs, which are comparatively more stable than mRNA-LNP systems, whose shelf-life is limited by the fragility of the mRNA cargo. While preliminary work demonstrates that the described formulation solutions can be applied to more sensitive cargoes, including mRNA, the benefits are dependent on ionizable lipid chemistry and are most pronounced with unsaturated lipids (Supplementary Fig. 18, Supplementary Table 8). As such, more comprehensive, long-term stability studies using these systems are necessary. Advancements in this field are expected to directly impact the shelf-life of these complex materials, convenience in their administration (e.g., improving in-use stability profile), or reduce the need for distribution of RNA-LNP drug products within a frozen cold chain to expand patient accessibility.

## Methods
### Materials and equipment
Chemical reagents were purchased from Avanti, Teknova Corning, or ThermoFisher and used without purification unless noted otherwise. D-Lin-DMA-MC3 was purchased from MedChem Express (Lot# 87108 and 218178), Organix (Lot# 569-146-1-2, 569-118-1), or MedKoo (Lot# YXM210629). ALC-0315 and ALC-0159 were purchased from Echelon Biosciences (lots E00288-156-15 and E00298-28-02, respectively). SM-102 was purchased from Echelon Biosciences (lot E00284-154-15). DODMA was purchased from Med Chem Express (lot 390274), DOTAP was purchased from Avanti (lot 890890C-200MG-B-180), and Dlin-KC2-DMA was purchased from Chem Scene (lot 607283). HPRT siRNA duplex (siHPRT, ID: XD-45798) was purchased from AxoLabs and has the following sequence: (i) sense strand: 5'-uscscuauGfaCfUfGfua-gauuuusasu-3'; (i) antisense strand: 5'-pasUfsaaaAfucuacagUfcA-fuaggasasu-3' (a, c, g, u: 2'-O-Methyl nucleotides, s: phosphorothioate; Af, Cf, Gf, Uf: 2'-Fluoronucleotides, p: (mono)phosphate. EPO mRNA (CleanCap EPO, SKU L-7209, Lot# WOTL76618) was purchased from TriLink Biotechnologies and has full length of 859 nucleotides with a proprietary sequence, and an open reading frame length of 582 nucleotides with the following sequence: AUGGGCGUGCACGAGUGCCCCGCCUGGCUGUGGCUGCUGCUGAGCCUGCUGAGCCUGCCCCUGGGGCCUGCCCGUGCUGGGCGCCCCCCCCCCGGCUGAUCUGCGACAGCCGGGUGCUGGAGCGGUACCUGCUGGAGGCCAAGGAGGCCGAGAACAUCACCACCGGCUGCGCCGAGCACUGCAGCCUGAACGAGAACAUCACCGUGCCCGACACCAAGGUGAACUUCUACGCCUGGAAGCGGAUGGAGGUGGGCCAGCAGGCCGUGGAGGUGUGGCAGGGCCUGGCCCUGCUGAGCGAGGCCGUGCUGCGCGGGGCCAGGCCCUGCUGGUGAACAGCAGCCAGCCCUGGGAGCCCCUGCAGCUGCACGUGGACAAGGCCGUGAGCGGCCUGCUGGGAGCCUGACCACCCUGCUGCGGGCCCUGGGCGCCCAGAAGGAGGCCAUCAGCCCCCCCGACGCCGCCAGCG

CCGCCCCCCUGCGGACCAUCACCGCCGACACCUUCCGGAAGCU-
GUUCCGGGUGUACAGCAACUUCCUGCGGGGCAAGCUGAAGCUGUA-
CACCGGCGAGGCCUGCCGGACCGGCGACCGGUGA.

Centrifuge and Eppendorf microcentrifuge tubes were purchased from Fisher Scientific. DI water was supplied via the Milli Q Integral 15 system. PD-10 column purification was performed via Cytiva Sepha-dexTM G-25 M columns (CAT# 17085101) and manufacturer gravity protocol. Bath sonication was performed using a Branson Bransonic® CPXH Digital Bath 1800. All heating was performed using a VWR Model 1224 water bath set to 50 °C. Masses for analytical measurements were taken on a Mettler Toledo XSR204 or XSE204 analytical balance. Mixing was performed with Teledyne ISCO 100DX syringe pumps, see Supplementary Fig. 1, or the NanoAssembler Ignite Plus mixing system. All nanoparticles were stored in 2 mL Schott Glass BT5933 vials with a 13 mm serum stopper and crimped with aluminum seals.

### RNA-lipid nanoparticle formation procedure

Lipid nanoparticles were formulated with the following procedure. First, D-Lin-MC3-DMA, cholesterol (Avanti), DSPC (Avanti), and DMG-PEG-2000 (Avanti) were removed from -80 °C storage and allowed to come to room temperature. In addition, siRNA powder was removed from -80 °C storage and allowed to come to room temperature. Lipids were then weighed out into glass scintillation vials. Cholesterol (55.8 mg) was placed in a vial, and 5.55 mL of EtOH was added to make a 10 mg/mL stock solution. The sample was then placed in a heated water bath. This was repeated for DSPC (29.6 mg lipid, 2.96 mL EtOH) and DMG-PEG-2000 (14.1 mg lipid, 1.40 mL EtOH). After approximately 10 minutes of heating, if any undissolved lipid remained, the solutions were then briefly sonicated. D-Lin-MC3-DMA (120.4 mg) was weighed into a glass scintillation vial, and 12.04 mL of EtOH was added. The solution was then mixed thoroughly without heating. Once all 10 mg/mL stock lipid solutions had been made, they were then combined in a 50 mL centrifuge tube and EtOH added for a final concentration of 12.5 mM lipid and a molar ratio of 50:38.5:10:1.5 (D-lin-MC3-DMA:Cholesterol:DSPC:DMG-PEG-2000). The siRNA stock solution was made by adding 1 mL of MilliQ H₂O to 10 mg of siRNA powder. Following this, 996.9 μL of siRNA stock solution was diluted to 90 mL with 0.05 M sodium citrate, pH 5 to make the siRNA working solution. Solution was mixed thoroughly, and siRNA concentration was measured on a Thermo Scientific NanoDrop One with a custom factor of 46.20. The lipid and siRNA working solutions were then loaded into the syringe pumps. Due to pump volume limitations, each batch was formulated as 2 × 60 mL batches and combined for a total volume of 120 mL. Samples were generated with a total flow rate of 12 mL/minute, 3:1 siRNA: lipid ratio, utilizing a T-mixing junction. Nanoparticles were collected in a tared 500 mL PETG media bottle. The first 2 mL, and last 1 mL of each batch were not collected. Each batch was diluted 1:1 with MilliQ H₂O or buffer prior to holding at 5 °C for 30 minutes. Dilution was performed on either the bulk sample after mixing or during mixing utilizing an in-line dilution. Sample weight was recorded prior to hold at 5 °C.

For ALC-0315 nanoparticles, the above procedure was followed with ALC-0315 (Echelon Biosciences) in place of D-Lin-MC3-DMA, and ALC-0159 (Echelon Biosciences) in place of DMG-PEG-2000. mEPO (CleanCap) was utilized as the cargo, with an N:P ratio of 6.0. mEPO working solution was prepared by diluting 593 μL of 1 mg/mL stock to 9 mL for a working concentration of 0.0659 mg/mL. Lipids were combined for a total lipid concentration of 8 mM, with a molar ratio of 46.3:42.7:9.4:1.6 (ALC-0315:Cholesterol:ALC-0159:DMG-PEG-2000). Total batch size was 12 mL, collected in a 15 mL Falcon tube. SM-102 nanoparticles were formulated using the same procedure, with SM-102 (Echelon Biosciences) in place of ALC-0315, and DMG-PEG-2000 in place of ALC-0159. Additionally, SM-102 nanoparticles were formulated with a molar ratio of 50:38.5:10:1.5 (SM-102:Cholesterol:DSPC:DMG-PEG-2000).

DOTAP (Avanti), DODMA (MedChemExpress), and DLin-KC2-DMA (ChemScene) were all formulated by following the above procedure with 10 mg/mL lipid stocks and siHPRT cargo to achieve a total lipid concentration of 12.5 mM, a total N:P ratio of 4.5, and a final molar ratio of 45:44:9:2 (Ionizable Lipid:Cholesterol:DSPC:DMG-PEG-2000). Nanoparticle samples were generated using the NanoAssembler Ignite Plus mixing system at a total flow rate of 12 mL/minute, 3:1 siRNA:lipid ratio, and a total batch size of 13 mL was collected for each ionizable lipid in separate 15 mL Falcon tubes.

### Degradation of MC3 lipid in ethanolic solution

MC3 lipid integrity within PBS 1X (Supplementary Fig. 6). MC3 (Organix, Cat# O-8640, Lot# AS-569-118-1) was solubilized in ethanol (200 proof, Supelco, Cat# EX0276-3, Lot# 62119) at a concentration of 4.00 mg/mL. Concentrated ethanolic solution was then pipetted into buffers to 25 vol% ethanol in either (1) PBS 1X (DPBS Modified, Cytiva, Cat# SH30028.02, Lot# AG29791191) or (2) 10 mM Histidine Buffer and mixed via repeated pipetting and vial inversion. Samples were then further diluted 1:1 vol% with ethanol (200 proof, Decon Laboratories, Cat# 2701, Lot# A08062105M), mixed thoroughly by vortex, visually inspected, and submitted for LC-MS analysis.

### Forced degradation sample preparation

To evaluate accelerated conditions, samples were spiked with either a solution containing 0.1 mg/mL Nickel (II) Chloride (Thermo Fisher), Copper (II) Chloride (Thermo Fisher), and Iron (III) Chloride (Thermo Fisher), or a 600 ppm peroxide stock solution. Metal spike solution was prepared by dissolving 10 mg of each of the respective metal salts into 100 mL of ultra-pure water. Peroxide stock solution was prepared by adding 2 mL of a 30% (w/w) hydrogen peroxide solution (Sigma-Aldrich) to a 100 mL volumetric flask and diluting to the mark for a final concentration of 600 ppm. Following this, 500 μL of LNP solution was added to a BT5933-type glass vial. Samples were then spiked with either 5 μL of the metal stock solution for a final concentration of 1 ppm, or 1.2 μL of the peroxide stock solution for a final concentration of 1 ppm. One set of samples was also vialed to serve as a control branch. Vials were then stoppered, crimped, placed in either a controlled 25 °C or 40 °C stability chamber, and pulled at either 2 weeks or 4 weeks. Individual vials were prepared for each respective time point.

### RNA-lipid nanoparticle buffer exchange via tangential flow filtration

All TFF processing was done utilizing a Repligen KrosFlo® KR2i TFF system equipped with a KrosFlo® KR Jr Auxiliary Pump for diafiltration, a feed scale, a permeate scale, and a KrosFlo® Back Pressure Control Valve. In addition, all samples were processed with a Repligen SIUS PDn 0.02 m2 (LP) PrS 100 kD cassette in a Millipore Sigma Pellicon® Mini Cassette Holder. Prior to nanoparticle processing, the TFF was first prepared by adding 250 mL of 0.5 N NaOH to the feed reservoir, and recirculating for 30 minutes. Following this, the system was emptied to waste, 250 mL of 0.1 N NaOH was added, and allowed to recirculate for 15 minutes. The 0.1 N NaOH was then processed to waste, and 150 mL of diafiltration buffer was added, and allowed to recirculate for an additional 15 minutes. The diafiltration buffer was then processed to waste, and the system was flushed with 500 mL of MilliQ H₂O. Samples were then retrieved from the 5 °C hold. Samples were either added directly to the tared TFF reservoir or filtered through a Pall VacuCap 0.8/0.2 μm filter prior to adding to the tared TFF reservoir. The sample weight in the reservoir was then entered into the "feed weight override" field of the Repligen TFF software. Samples were then processed at a flow rate of 36 mL/minute, TMP 5, a target concentration factor of 13.4, and 6x diafiltration volumes. After TFF processing, air was pumped through the TFF system, and samples were decanted into a 50 mL centrifuge tube. Following this, samples were filtered through a Millipore Sigma 0.22 μm PES Steriflip Vacuum Filtration System in a biosafety cabinet. Samples were then

aliquoted into 2 mL Schott Glass BT5933 vials with a 13 mm serum stopper and crimped with aluminum seals with an Eppendorf Repeater® E3x repeater pipette. Each vial was filled with 1 mL of filtered LNP. Samples were then segregated into fiberboard storage boxes (Thermo Scientific) by storage condition, and stored at either 2–8 °C, or at room temperature (22–25 °C). After sample testing, the remaining LNP solution was stored at -80 °C until LC/MS analysis was able to be performed. In some instances, at a small scale (Fig. 4 and Table 1), buffer exchange was performed via a PD-10 desalting column (Cytiva cat#17085101). Buffer exchange was performed via manufacturer recommended procedure.

## General lipid nanoparticle analysis procedure

Size analysis: The LNP solution was diluted 50x in PBS 1X (10 μL LNPs in 490 μL PBS 1X) in a ZEN0040 disposable cuvette. Size was analyzed with a Malvern Zetasizer Nano dynamic light scattering. Samples were measured with the liposome default setting (refractive index of 1.45, absorption 0.001) and dispersant set to PBS (refractive index of 1.33, viscosity of 0.8872 mPa.s). Temperature was set to 25 °C with an equilibration time of 30 seconds. Instrument default settings were used for all other settings. Nanoparticle tracking analysis: Nanoparticle tracking analysis (NTA) measurements were performed on a NanoSight NS300 instrument (Malvern Panalytical) equipped with a scientific CMOS camera, and green laser module (532 nm), and NTA version software 3.4. LNP samples were diluted 1000x fold in modified DPBS (Cytiva) and loaded into a 1-mL disposable syringe. A syringe pump was used to flow the samples into the instrument chamber, and a constant syringe pump speed of 35 (arbitrary units) was used during data collection. Samples were measured for 30 seconds, repeated 5 times ($n = 5$). The detection threshold in the NTA software was set to 6, and the maximum jump distance and the minimum track segment length were both set to auto.

## Lipid or LNP LC-MS Analysis

LC, Thermo Vanquish or Agilent 1290 Infinity II UHPLC system; MS Thermo Orbitrap EQ HF-X, Exploris 240™ or Eclipse™ Tribrid™ mass spectrometer; Column, Waters Acquity UPLC BEH 300 A C4, 2.1×100 or 150 mm, 1.7 mm column; Mobile phase A, 0.05% TFA in H₂O, B, 0.04% TFA in acetonitrile for lipid analyses or Mobile phase A, 10 mM DIPEA, 100 mM HFPA in H2O, B, acetonitrile for siRNA or related analyses. Aliquot of each lipid or LNP samples was directly subjected to LC-MS analysis or diluted with ethanol (1:2 to 1:4) before LC-MS analysis. For lipid analysis, the column was equilibrated at 0% B at 50 °C and flowrate of 0.25 mL/min. After the injection, the column was hold at 0% B for 2 min, rapid increasing to 65% B in 1 min, gradually to 75% B in 6.5 min, to 100% b in 1 min, hold at 100% B for 2.3 min, back to 0% B and hold at 0% B for several minutes before the next injection. The mass spectrometry was operated in positive mode with service scan, Orbitrap Resolution of 240000, Scan Range (m/z) of 120 to 1000; tandem mass, Orbitrap Resolution of 15000, HCD Collision Energy (%) of 30, Ion Charge State for MS2 of 1-2, Cycle Time of 1 sec, Dynamic Exclusion of 5 sec. All the parameters were standard or auto. For siRNA or related analysis, the column was equilibrated at 5% B at 70 °C and a flow rate of 0.3 mL/min. After the injection, the column was hold at 5% B for 2 min, increasing to 11% B in 1 min and the same decreasing flowrate to 0.2 mL/min, to 13% B in 7 min, to 95% B in 5.5 min 100% b in 1 min, hold at 100% B for 2.5 min, back to 0% B and hold at 0% B for several minutes before the next injection. The mass spectrometry was operated in negative and positive mode with Orbitrap Resolution of 240000, Scan Range (m/z) of 400–2000 (negative) and 500–6000 (positive), Source Fragmentation Energy of 25 V (negative) and 100 V (positive). Negative ions of siRNA and related impurities were selected for target tandem MS analysis with step HCD Collision Energy (%) of 25, 29, and 33, Orbitrap Resolution of 30000, RF Len (%) of 80, Normalized AGG Target (%) of 1000, and Maximum Injection Time (ms) of 800. All the parameters were standard or auto.

## Quantitative analysis of LC-MS analysis

Peak area of each intact lipid, siRNA, and their impurities identified based on tandem MS was obtained with Thermo Quan Browser software. For lipids, top two or three ions of each molecule with a narrow m/z range (the range might be various according to ion resolutions) were selected to get the extracted ion chromatography (XIC or EIC) peak and its area. For siRNA, all ions of each molecule were selected to get the extracted ion chromatography (XIC or EIC) peak and its area. The relative percent of each lipid or siRNA impurity was calculated after all peak areas of its related molecules were obtained.

## RiboGreen analysis

Two standard curves of siRNA were prepared, one curve of siRNA in 1X TE buffer pH 5 (Prepared from Invitrogen 20x TE buffer Cat# R11490), and one of siRNA in 0.1% Triton X-100 in TE (Prepared from Sigma Aldrich Triton X-100). To start, 20 μL of 0.2 mg/mL siRNA was added to 980 μL of 1X TE or 0.1% TX and mixed well for a 4000 ng/mL standard. Following this, 250 μL of the 4000 ng/mL standard was taken and serially diluted 1:1 to a concentration of 8 ng/mL. A blank was also prepared for each standard curve. Following this, samples were prepared. First, 10 μL of LNP was added to 90 μL of 1X TE and mixed thoroughly to make Sample Dilution 1. Following this, 10 μL of Sample Dilution 1 was added to 990 μL of 1X TE or 0.1% Triton X-100 in TE and briefly vortexed. After samples were prepared, 100 μL of the standard curves and samples were pipetted in duplicate to a 96-well plate (Fisherbrand Cat# 12565501). Ribogreen dye (Invitrogen Cat# R11491) was diluted 200x in TE buffer, mixed well, and 100 μL of diluted dye was added to each well of the 96-well plate. Immediately following this, samples were measured using a TECAN Spark 10 M plate reader with 480 nm excitation wavelength, and 520 nm emission wavelength.

## MFI analysis

MFI analysis was carried out on a ProteinSimple MFI 5200. Immediately prior to analysis, LNP samples were diluted 100x in 1x PBS pH 7.4 and mixed by gentle inversion. Samples were analyzed using the MFI View System Software. The analysis method was set with a purge volume of 0.2 mL and an analyzed volume of 0.6 mL. Particles greater than or equal to 1 μm in diameter, and smaller than or equal to 100 μm were reported.

## Relative turbidity

Relative turbidity measurements were taken utilizing a C Technologies SoloVPE. Samples were analyzed using the "Quick Survey" function and a pathlength of 5 mm. Absorbance at 400 nm was recorded and converted to % transmittance utilizing the equation %T = $10^{2-A}$, where %T is the % transmittance and A is the absorbance at 400 nm.

## Osmolality measurements

Osmolality measurements were taken using a Precision Systems Osmette III utilizing the manufacturer's protocol.

## pH measurement

pH measurements were taken using a Mettler Toledo SevenCompact pH meter. Calibration was performed day of measurement utilizing a 4-point calibration.

## HPLC Analysis

HPLC analysis of siRNA was carried out on an Agilent 1290 Infinity UPLC equipped with a UV-Vis detector. Column, Waters Acquity™ UPLC Peptide BEH C18 300 A 2.1 × 150 mm 1.7 μm. Mobile Phase A, 76 mM HFIP, 28 mM di-n-butylamine with 10 μM EDTA in water. Mobile Phase B, 80% Methanol, 10% Isopropanol, 10% Water. Prior to analysis, samples were extracted by adding 100 μL of LNP to 900 μL of 60 mM ammonium acetate in isopropanol. Samples were centrifuged at 18,000 g for 30 minutes at a temperature of 5 °C. The supernatant was

then pipetted off, and 1000 μL of isopropanol was added. The samples were then spun again, the supernatant pipetted off, and the samples were allowed to dry under vacuum. Once dried, the samples were reconstituted with 100 μL of ultra-pure water and transferred to HPLC vials for analysis.

## Computational details and methodology: 1. Density functional theory

All computational calculations are performed using the Gaussian 16 program[40]. Geometrical structures are optimized at the M062X/6-31 + G(d,p) level of theory[41]. Vibrational frequency calculations are performed on the optimized structures at the M062X/6-31 + G(d,p) level of theory to verify the optimized geometries are actual minima on the potential energy surface; no negative frequencies are observed. Bond dissociation energies (BDE) are calculated as the reaction enthalpies of C-H (shown in red color in Fig. 4A) bond homolysis reaction[42]. The Gibbs free reaction energies shown in Supplementary Fig. 13 are calculated by subtracting the free energies of reactants from that of the products. Atomic coordinates of the optimized computational models are included within Supplementary Data 1–8 as plain, unformatted text files. 2. Molecular dynamics (MD) simulation: The all-atom molecular structure of double-stranded siRNA was generated using the 3D Builder function in Schrödinger Maestro[43]. Since the modified RNA structure could not be generated directly in Maestro, chemical modifications were applied manually within the software. All-atom structures of positively charged MC3 and its oxidative byproduct were also generated in Maestro. The double-stranded siRNA, MC3, and MC3 byproducts were minimized in Maestro. The components of our model system were assembled using Packmol[44] in a cubic simulation box with each side set to 9 nm. Periodic boundary conditions were applied. Two systems were built for comparison: one containing siRNA paired with a single MC3 molecule, and the other with siRNA paired with a single MC3 byproduct molecule. Both systems were solvated using the TIP3P water model[45] in 1X PBS to replicate experimental conditions. The CHARMM36 force field[46], including nucleic acids[47] and its extension for modified nucleotides[48], was used to model our systems model the modified siRNA. The CHARMM General Force Field[49,50] was employed to parameterize the cationic ionizable lipids[51]. Simulations were performed using the GROMACS 2023 package[52]. The LINCS algorithm was used to constrain bonds involving hydrogen atoms[53]. Long-range electrostatic interactions were handled using the particle mesh Ewald (PME) method[54] with a real-space cutoff of 1.2 nm. Van der Waals interactions were treated with a force-switch modifier, with a cutoff radius of 1.2 nm and a switch function applied over the range of 1.0–1.2 nm. Minimization, equilibration, and production MD simulations were performed on three independent systems. Energy minimizations were carried out using the steepest descent algorithm for 5000 steps. Following minimization, equilibration was conducted in an NpT ensemble with a 1 fs time step for 200 ns. Temperature coupling was achieved using the Nose-Hoover thermostat[55,56], with separate coupling groups for the solute and solvent, maintaining the system at 303.15 K with a coupling time constant of 1.0 ps. Initial velocities were generated according to a Maxwell-Boltzmann distribution at 303.15 K. Production runs were performed using the leapfrog integrator[57] with a 2 fs time step over 500,000,000 steps, corresponding to 1 μs of simulation time. The Parrinello-Rahman barostat[58] was used during the production phase to maintain the system at a reference pressure of 1.0 bar, with a coupling time constant of 5.0 ps and a compressibility of $4.5 \times 10^{-5}$ bar$^{-1}$. Radial distribution functions (RDF) were calculated using the GROMACS utility "gmx rdf," and simulation snapshots were visualized with VMD[59].

## RT qPCR: 1. Cell collection and cDNA synthesis

HeLa cells were obtained from the American Type Culture Collection (ATCC). The cell line was authenticated by ATCC using short tandem repeat (STR) profiling. No additional authentication was performed within our laboratory. The cell line was routinely tested and confirmed to be free of mycoplasma contamination.HeLa cells are listed in the ICLAC register of misidentified cell lines. However, their identity and use in this study are appropriate and intentional, as HeLa is a widely used and well-characterized model cell line. HeLa cells were seeded on Day 0 and transfected on Day 1. Sample delivered by Lipofectamine™ RNAiMAX transfection reagent was used as a positive control. After 24 hours of forward transfection, cells were subjected to the reverse transcription process using the TaqMan™ Fast Advanced Cells-to-CT™ Kit. cDNA was synthesized following the manufacturer's protocol. The resulting cDNA was stored at -20°C until further use. 2. Quantitative PCR (qPCR): qPCR was conducted on the QuantStudio 5 system using the TaqMan Fast Advanced Master Mix and TaqMan qPCR assays. Each 20 μL reaction mixture contained 10 μL of TaqMan Fast Advanced Master Mix, 0.33 μL of 60x TaqMan qPCR assay for both the target and reference genes, 2 μL of cDNA, and 7.34 μL of nuclease-free water. Thermal cycling began with a UDG activation step at 50°C for 2 minutes, followed by a polymerase activation step at 95°C for 1 minute. This was followed by 44 cycles of denaturation at 95°C for 10 seconds and annealing/extension at 60°C for 30 seconds. A dissociation curve step was added to confirm the specificity of the PCR products, ramping from 65°C to 95°C. 3. Data Analysis: Relative gene expression levels were quantified using the comparative 2^(-ΔΔCT) method, normalized to the housekeeping gene bActin. Results were expressed as a percentage change relative to control samples. Reactions were performed in duplicate to ensure the reproducibility and accuracy of the results.

## Statistical analysis

To assess whether differences in two samples at a given time point were statistically significant (e.g., size of RNA-LNPs immediately after buffer exchange into either PBS or Histidine), a two-tailed Student's t-test assuming an unequal sample variance was used. To assess whether differences in samples over an extended time were statistically significant (e.g., changes in particle size over time at a given temperature), data were evaluated by linear regression analysis, and two-tailed ANOVA p-values were reported. Data analysis was performed using JMP Version 17.2.0.

## Inclusion & ethics statement

All authors have fulfilled the criteria for authorship. This research has included local researchers throughout the process within study design, study implementation, data ownership, intellectual property, and authorship. Roles and responsibilities were agreed among collaborators ahead of the research.

## Reporting summary

Further information on research design is available in the Nature Portfolio Reporting Summary linked to this article.

# Data availability

All processed data supporting key conclusions are available within the article and the Supplementary Information. All data underlying this study are available from the corresponding author upon request. Source data are provided with this paper.

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

## Acknowledgements

The authors dedicate this work to the memory of Young-Ho Song, whose valuable feedback on the data, study design, and project direction was instrumental to this work and whose scientific curiosity, generosity, and kindness continue to inspire us. This study was entirely funded by Eli Lilly and Company, Indianapolis, IN. Cryo-electron microscopy data were obtained via contract with the University of Massachusetts Chan Medical School. The authors thank Mindy Blair Forst and Adam McFarland for supporting [1]H-NMR data analysis (Supplementary Fig. 12). They thank Scott Alan Frank for supporting the DFT modeling of the MC3 oxidation pathway (Fig. 4A, Supplementary Fig. 13). They thank Scott Brown for assisting in modeling non-covalent interactions between lipids and RNA molecules of interest (Supplementary Fig. 15). They thank Zhefeng (Eddie) Li and Zhe Tan for supporting bioassay data (Fig. 5). They thank Byungkook (Qook) Lee for feedback on data and study design.

## Author contributions

D.E. was responsible for writing the manuscript, which was edited by all authors and select contributors. L.H., Z.Y., and T.W. identified MC3 lipid instability (Supplementary Fig. 6), with T.W. providing study design, Z.Y. carrying out experiments, and L.H. identifying degradants by LC/MS. D.E., O.L., D.C., and T.W. bridged findings to MC3-based RNA-LNPs, with D.E., O.L., and D.C. executing experiments and D.E. providing study design. D.E. worked to elucidate the structure of MC3 degradants, the oxidation mechanism, and identified electrophilic oxidant degradants as a root cause of RNA-lipid adduct formation with experimental assistance from L.H., J.B., F.B.S., D.C., K.R., N.W., and S.C. Expanding the scope with other ionizable lipids (DOTAP, DODMA, DLin-KC2-DMA) was done by D.E., L.H., D.C., K.R., N.W., and S.C. (Supplementary Figs. 9–11). J.B. was responsible for NMR data collection, analysis, and dienone structure elucidation (Supplementary Fig. 12). L.H. executed all LC/MS experiments (including structure analysis and RNA-lipid adduct identification) for experiments throughout the manuscript. X.L. and H.J. were responsible for executing the *in cellulo* bioassay data (Fig. 5). F.B.S. executed DFT modeling of MC3 and the oxidation pathway (Supplementary Fig. 13). Y.O. performed molecular dynamics simulations on the lipid-RNA interactions (Supplementary Fig. 15). G.N. and D.Y. executed nanoparticle tracking analysis to support the mechanism of colloidal aggregation (Supplementary Fig. 4). T.W. provided project direction input throughout. All authors have given approval to the final version of the manuscript.

## Competing interests

At the time of the manuscript preparation all authors in this report are, or were previously, employees of Eli Lilly and Company, Indianapolis, IN. D.E., L.H., O.L., Z.Y. and T.W. have filed intellectual property related to this manuscript (including Patent No. WO2024137423A1, published but pending). D.E., L.H., D.C., Z.Y., F.B.S., J.B., Y.O., X.L., H.J., K.G.R., G.N., D.Y., and T.W. own stock in Eli Lilly and Company. N.W. and S.C. declare no competing interest.
