## [Transparent Peer Review file · Nature Communications]

Buffer Optimization of siRNA-Lipid Nanoparticles Mitigates Lipid Oxidation and RNA-lipid Adduct Formation

Corresponding Author: Dr Daniel Estabrook

Version 0:

Reviewer comments:

Reviewer #1

(Remarks to the Author)

The Impact of Buffer in RNA-LNP Formulations: Mitigating Oxidation of Ionizable Lipid Tails and RNA-Lipid Adduct Formation by Estabrook and coworkers addresses an important topic within pharmaceutical mRNA nanoparticle development field.

The authors study various stability limiting phenomena, including oxidative degradation. Although this is an important topic, which is relevant for many scientists in practical pharmaceutical development, it contains only very little new information and also no new insight into the mechanisms of degradation from a scientific perspective.

The paper is very lengthy and may be more seen as a review than a report on novel discoveries.

Basically nothing in this paper is new. Some examples:

The authors write:

- “Herein, we demonstrate that the room temperature stability of MC3-based siRNA-LNP can be improved by inclusion of mildly acidic, antioxidant-containing buffers”.

This is not new and can be found, for example, in publicly available patents from players like Moderna and BioNTech.

- “We show that while phosphate-based formulations limit the room temperature stability of siRNA-LNPs to two weeks, the use of a histidine-containing buffer allows for room temperature stability of 6 months to date”.

Also this has been described and patented by other players in the mRNA field.

- “The stability of these nanomaterials is shown to be correlated with the oxidation of the unsaturated lipid tail, which results in the production of a dienone species. We show that these oxidative byproducts have three deleterious effects, namely they (i) introduce conformational changes in the lipid tail, (ii) produce hydrogen bond acceptor that non-covalently interacts with RNA, and (iii) generate an electrophilic species that react with nucleophilic residues in RNA cargo to produce RNA-lipid adducts”.

This is known to developers in the field since decades ago solutions can be found in the respective patents.

More specific comments:

This is interesting but this phenomenon is specific for unsaturated lipids such as DLin-MC3-DMA since the alkene groups are prone to oxidize over the time.

1. Oxidation of ionizable lipid through amine of the lipid followed by adduct formation has been predicted previously (Ref 2). Did author notice such phenomenon with siRNA and MC3 lipid?
2. Did authors investigate other saturated lipids such as Pfizer/Moderna COVID-19 lipids and siRNA? Can the proposed mechanism work with histidine buffer?
3. Figure 5, siRNA functionality, did authors evaluate the activity of LNPs in other buffers reported in table 1?
4. Figure S13, why one lipid tail involved in adduct formation with siRNA?

In summary, the manuscript should be rejected for publication in Nature Communications. It may be suitable for a more specialized journal, as a kind of review supported by the group own data.

Reviewer #2

(Remarks to the Author)

The manuscript by Estabrook et al. focuses on demonstrating that lipid nanoparticle (LNP) formulations containing MC3 lipids could react with RNA cargoes. The suggested mechanism is the oxidation of MC3 lipid tails resulting in production of a dienone species and generating an electrophilic degradant that reacts with RNA to form lipid-RNA adduct. The authors also suggest modifications to the LNP formulation that may remedy this problem.

There is much interest in both making the LNP formulations more efficient and shed light on the mechanisms behind instability and inefficiencies of the existing formulations and thus this manuscript deals with a significant issue. On the other hand, there are major issues with the manuscript that need to be addressed. I will focus on the computational side of the work given my own expertise.

The main issue with the computational side of this work is the lack of clarity of the methodology. There is very little information provided on the details of both the DFT and MD methods. For instance, it is not even stated how long are the MD simulations and how many times they have been repeated and how the systems were set up. Without knowing the basic details of the simulations and analysis techniques it is not possible to determine whether the conclusions are sound or not.

Another issue is the use of CGenFF for the lipids and RNA forcefield. It is strange that the authors have used a generic force field for the most important part of the simulation while there are more specific forcefields for lipids and RNA.

Based on these observations the computational part of this manuscript is not acceptable in my opinion.

Reviewer #3

(Remarks to the Author)

Reviewer Comments

Thank you to the authors for sharing this interesting work and for investigating how oxidation of unsaturated ionizable lipids can contribute to the loss of RNA LNP potency. In general, the study is interesting and the chemical analysis was interesting. However, there are no statistical analyses in this study; these tests are needed to help support some of the major interpretations of key data. Additionally, a few controls should be added. Please consider addressing the comments below to improve this manuscript.

Line 133: beyond citing it is the likely buffer of Onpatro, please comment on why the calcium and magnesium would be omitted.

Line 146-158: the authors say that nanoparticles were aggregating and perform measurements by MFI to show an increase in the number of particles that are $\geq 1\mu\text{m}$. The authors said that sub 100 nm particles were aggregating. In its current form, I don't think the presented data supports this claim. The DLS data does show a decline in intensity over time of the nanometer-scale nanoparticles, but intensity values aren't necessarily comparable between conditions. If the DLS used a different level of attenuation, would this make intensity numbers incomparable? I suggest the authors measure nanoparticle distribution by an orthogonal method, such as the NanoSight machine and check for a decline in the sub 100 nm population. Once done, please revise these statements accordingly.

Line 159 – 162 and Figure 2: please perform the appropriate statistical tests to show that the EE and RNA content remained the same.

Line 191-193: can the authors please show the LC/MS of their starting MC3 materials (MC3 in solution and not formulated into an LNP), to determine if it arrived oxidized or hydrolyzed? Also, separately oxidize and hydrolyze samples of MC3 and show the LC/MS. These data should be included as positive controls.

Line 212-213: please clarify or better support the supposition that oxidation likely happens within the LNP core. Is the idea that the core contains MC3 that would react with the RNA? Its ok to be speculative, but please describe this spatial hypothesis in better detail.

Line 214-216: These statements on concentration-dependent oxidation don't appear to be supported by statistical analysis of the Figure S6 data. Please increase the number of replicates for S6 and perform the need statistical analysis. Alternatively, reconsider the inclusion of these statements. Perhaps add in an oxidizing agent to create a positive control for oxidation, which would benchmark the loss of potency (change in IC50) that is caused by oxidation.

Lines 230-233 and Figure 3: no statistical analysis was done to support the statement that sizes and PDI were unchanged. Please address this issue.

Line 383-385: Figure 5B shows error bars, but no statistical analysis is done. Also, the statement says that room temperature LNPs in histidine buffer showed no loss in potency; however, the Figure 5B table shows everyone to be within 0.01 of each other. Please reconsider these statements after statistical analysis.

Figure S2: please clarify if this was N=1, or an average of a triplicate measurements, etc.

Figure S6: Indicate the number of replicates.

Figure S9B: difficult to see axis labels. When one zooms in, it is fuzzy and one cannot make out the numbers.

Figure S12A: Thank you for providing the sequence. However, please indicate what upper and lower case letters mean for the siRNA, and what the modification abbreviations mean.

Figure S13: the LC/MS traces are very low resolution and difficult to appreciate; they become fuzzy when one zooms in.

Tables S1 to S4. The nomenclature and shorthand used in the leftmost column and header rows aren't fully explained. Either rename them or label them more clearly. For example, in the Table S4 "siRNA Degradation" column, what does each row entry mean?

Version 1:

Reviewer comments:

Reviewer #2

(Remarks to the Author)

The authors have addressed my concerns.

Reviewer #3

(Remarks to the Author)

Thanks for working to extend the study and address all previous reviewer comments. I am unable to comment on the simulation work, so focused on comments from Rev 1 and Rev 3.

For Rev 3, given the retraction and attenuation of the conclusions, this is probably not suitable for Nature Communications.

- the concentration-dependent data was compelling, but now it has been retracted due to experimental limitations
- the authors claim that smaller ~70 nm particles are aggregating over time, causing a loss in this population. however, we cannot see an increase in larger particles, and the authors say no meaningful count of particles between 300–1000.
- Thanks for incorporating statistical analysis of Figure 2 and other figures. However, the figure captions must be updated to include statistical analysis info (not just the main text).

Seeing the feedback from Rev 1 and the rebuttal, the work should probably test more unsaturated lipids to show this phenomena is broadly applicable, and the degree to which it is a problem.

- There are no methods around the production of the SM-102 and ALC-315 LNPs. Please ensure the sequences of all RNA used, including mRNA, are reported.
- As a control, testing unsaturated lipids make sense. But to increase impact, showing that this mechanism applies to all unsaturated lipids would be of interest; otherwise, the impact feels limited the data generated by MC3 only.

The big idea is that unsaturated tails are an issue, but without verifying this in other configurations (e.g. would an unsaturated bond that is sterically hindered cause the same issues, or must it be in a chain that is accessible?). The work's impact would increase if verified across other (FDA approved or not) unsaturated lipids. The authors mentioned that new libraries have unsaturated lipids in them. Trying an extra two from these libraries could be great.

Version 2:

Reviewer comments:

Reviewer #4

(Remarks to the Author)

I was asked to provide a 4th review opinion to this manuscript, considering the previous rounds of review and focusing on Reviewer 3's comments.

In short, the authors have addressed prior concerns well, and I recommend to accept the paper for publication.

However, I do have a suggestion. Since the study focuses on siRNA, it would be helpful to mention in the discussion and/or conclusions potential limitations of the study, namely that the findings are most closely related to stability of short, double-stranded siRNA containing LNPs. The data on siRNA should, but may not, completely translate to longer, single-stranded mRNA. Since the introduction and main text heavily notes mRNA LNP vaccine instability as justification, readers will need to understand that the data was accumulated mainly for siRNA. I assume readers will implement this finding on buffers for various mRNA LNPs in the future, so it will be important for them to understand potential limitations as they further

investigate these important findings.

The authors may also consider changing the general "RNA" text in the titles, section headings, and main text to the more specific "siRNA" text since siRNA is the main cargo examined in this paper (other than a small subset of mRNA data in Figure S18 that may be insufficient to establish generalizability).

This study is highly relevant and well executed. Indeed, the benchmark formulation here is an siRNA LNP that mirrors FDA-approved Onpattro. So the data is good, and I just think readers may need to understand more clearly the work in context of the broad spectrum of RNAs under pre-clinical and clinical development (e.g., siRNA, miRNA, mRNA, tRNA, mRNA, sgRNA, pegRNA, saRNA, circRNA, etc.), where it seems the major finding relates to data describing siRNA.

Overall, the topic and results are compelling and suitable for publication.

“The Impact of Buffer in RNA-LNP Formulations: Mitigating Oxidation of Ionizable Lipid Tails and RNA-Lipid Adduct Formation”.

Point-by-point responses by the authors (in red) to the reviewers' comments, reproduced verbatim.

Reviewer #1 (Remarks to the Author):

The Impact of Buffer in RNA-LNP Formulations: Mitigating Oxidation of Ionizable Lipid Tails and RNA-Lipid Adduct Formation by Eastbrook and coworkers addresses an important topic within pharmaceutical mRNA nanoparticle development field.

The authors study various stability limiting phenomena, including oxidative degradation. Although this is an important topic, which is relevant for many scientists in practical pharmaceutical development, it contains only very little new information and also no new insight into the mechanisms of degradation from a scientific perspective.

We thank the Reviewer for their comments and apologize for any ambiguous language regarding the novelty (or lack thereof) of particular concepts addressed in the manuscript. The Reviewer is of course correct that certain concepts discussed within the manuscript, like the issue of oxidative degradation of lipids (including MC3), are well-known, and we have revised the manuscript to make this discussion more clear. However, to our knowledge the following concepts are not known within the literature:

- i. the primary degradant discussed within the manuscript: the E,Z-dienone of MC3
- ii. any analogous mechanism for degrees of unsaturation within ionizable lipids in the context of RNA-LNPs
- iii. the mechanism proposed for RNA-lipid adduct formation, or
- iv. the use of histidine-containing buffers to mitigate these undesired transformations and allow for room temperature storage of RNA-LNPs

We respectfully disagree with the Reviewer that elucidation of this mechanism provides no new insight into pharmaceutical development of RNA-LNPs, as it contributes a novel pathway for RNA-lipid adduct formation (the second to our knowledge, in addition to Packer *et al.*^[13]) and provides explanatory power and a mechanism for why the addition of antioxidants mitigates these adducts.

The paper is very lengthy and may be more seen as a review than a report on novel discoveries. Basically nothing in this paper is new. Some examples:

The authors write:

- “Herein, we demonstrate that the room temperature stability of MC3-based siRNA-LNP can be improved by inclusion of mildly acidic, antioxidant-containing buffers”.

This is not new and can be found, for example, in publicly available patents from players like Moderna and BioNTech.

We are unable to find any prior art that negates this claim. We would greatly appreciate if the Reviewer could provide a specific reference to the invention he/she is referring to. However, to address the Reviewer's concerns, we revised the manuscript to clarify that

the novelty of the manuscript does not lie within merely adding antioxidants to RNA-LNP formulations, but rather primarily in describing (i) the novel oxidized degradant and its role in RNA-lipid adduct formation, and (ii) the use of histidine-containing buffers to mitigate these undesired transformations. These changes are summarized below:

Submitted manuscript	Revised manuscript
Herein, we demonstrate that a common benchmark formulation in the field is prone to oxidation of unsaturated hydrocarbons within the ionizable lipid tail. This oxidation results in production of a dienone species that changes the conformation of the lipid tail...	Herein, we demonstrate that the oxidation of unsaturated hydrocarbons within ionizable lipid tails results in production of a dienone species that changes the conformation of the lipid tail...
“matrices, including mildly acidic, antioxidant-containing histidine formulations”	“ matrices, including mildly acidic, histidine-containing formulations”
“To our knowledge, this is the first demonstration of antioxidant-containing formulations showing such a dramatic impact on improving RNA-LNP room temperature stability.”	“To our knowledge, this is the first demonstration of histidine-containing formulations showing such a dramatic impact on improving RNA-LNP room temperature stability.”

- “We show that while phosphate-based formulations limit the room temperature stability of siRNA-LNPs to two weeks, the use of a histidine-containing buffer allows for room temperature stability of 6 months to date”.

Also this has been described and patented by other players in the mRNA field.

To the best of our knowledge, this has not been described elsewhere. We would greatly appreciate if the Reviewer could provide a specific reference to the invention they are mentioning. We are only aware of two related patents, both of which particularly pertain or frozen or freeze-dried formulations, and not solution phase drug products: (i) BioNTech’s “Pharmaceutical compositions comprising particles and mRNA and methods for preparing and storing the same” (WO2022101486A1) and (ii) GlaxoSmithKline’s “Freeze-drying of lipid nanoparticles encapsulating RNA and formulations thereof” (WO2023021427).

- “The stability of these nanomaterials is shown to be correlated with the oxidation of the unsaturated lipid tail, which results in the production of a dienone species. We show that these oxidative byproducts have three deleterious effects, namely they (i) introduce conformational changes in the lipid tail, (ii) produce hydrogen bond acceptor that non-covalently interacts with RNA, and (iii) generate an electrophilic species that react with nucleophilic residues in RNA cargo to produce RNA-lipid adducts.”

This is known to developers in the field since decades an solutions can be find in the respective patents.

While each of these three general phenomena have been described previously for other systems, the claim quoted here is specifically referring to oxidation of unsaturated lipid tails resulting in a reactive E,Z-dienone byproducts described within the manuscript. As

previously mentioned, to the best of our knowledge this degradant has not been described in the context of RNA-LNPs, nor has a drug product matrix strategy for mitigating it. We demonstrate this mitigation is of significant, practical value, as understanding interactions between excipient degradants and active pharmaceutical ingredient is necessary to optimize pharmaceutical drug products. To make a holistic rebuttal, we would appreciate if the Reviewer could provide citations for the patents and solutions that he/she believes make this work derivative. Towards these ends, we have provided a Table below that differentiates the claims made within the paper from those we are aware of within the field:

Previously described phenomena	Specific claim and citation	Novelty of the manuscript
Use of antioxidants in RNA-LNPs.	[1] Moderna: Antioxidants are included as a broad formulation claim. Shown to be compatible with formulation, but no clear benefit to either LNP size or (m)RNA purity over 3 days at 40 °C. Optimal formulations (DTPA as chelator) unstable at temperatures above 2–8 °C. [3-4] Moderna: Antioxidants and chelators are included as a broad formulation claim. No data provided. [5] Protiva: Specifically claims use of EDTA as a metal chelator. Improved particle stability for EDTA, but not citrate. Reduced PS-to-PO conversion using EDTA at 5 °C or RT. Encapsulation efficiency cannot always be retained at RT. Lipid degradation byproducts uncharacterized. [6] Alnylam: Antioxidants are included as a broad formulation claim for MC3 siRNA-LNPs. No data is provided.	- A wide number of particle attributes are specifically shown to be improved using antioxidants, as opposed to be captured within a broad formulation claim. Improved attributes include:  (i) Particle size (ii) Particle polydispersity (iii) Intact ionizable lipid & identified degradants linked to (v) (iv) Intact RNA cargo (v) Mitigated formation of RNA-lipid adducts (vi) In cellulo drug product potency
Use of histidine to improve RNA-LNP stability	[7] Manuscript: Use of histidine buffer, but to improve mRNA drug substance stability (no use of nanoparticle carriers). [2] Histidine leveraged to increase transfection efficiency, no benefit	- Use of histidine within siRNA-LNP drug product matrix development to improve stability of solution product at refrigerated or

	to stability of drug product claimed or demonstrated. [8] BioNTech: Use of histidine buffers for frozen or freeze-dried RNA-LNPs. [9] GlaxoSmithKline: Use of histidine buffers for freeze-dried self-amplifying mRNA-LNPs.	even room temperature (i.e., 4–22 °C).
[Lipid oxidation] results in conformational changes in the lipid tail and produces hydrogen bond acceptors that non-covalently interacts with RNA	[10,11] Known that oxidation of unsaturated bonds, including those within linoleic acid tails found in the byproduct described herein, can result in conformational changes. Not presented in the context of LNPs. It is known that carbonyls act as H-bond acceptors. The significance of H-bonding in ionizable lipid design has been investigated previously (e.g., [12]), though primarily within the head group.	Claims are being made specifically for the disclosed E,Z-dienone byproduct of MC3, and the impact such byproducts have in an RNA-LNP system (not just free lipid in solution).
[Lipid oxidation] generates an electrophilic species that react with nucleophilic residues in RNA cargo to produce RNA-lipid adducts	[13] Moderna authors described N-oxidation results in lipid aldehyde generation and subsequent mRNA-lipid adduct formation.	Describes a distinct electrophilic oxidized degradant, and the novel pathway that results in RNA-lipid adduct formation. Thus, this work represents the second example of the concept, following Packer et al.'s work and reports within lipid metabolism that supports its formation [14]
Additional citations from relevant, recent publications highlighting that oxidation is an ongoing challenge for RNA-LNPs	(i) AAPS PharmSciTech 2022, 23, 151. "Few publications offer pharmaceutically relevant and aggregated information on the oxidative stability of such complex lipid formulations [such as solid lipid nanoparticles]." (ii) Int. J. Pharm. 601, 2021, 120586. "Other excipients that could be added are antioxidants, non-reducing free radical scavengers (e.g., ethanol) or metal chelators (Evans et al., 2000). However, the question remains to what extent they indeed ameliorate the stability of mRNA-LNP formulations during storage below or above 0 °C." (iii) Polymers 2022, 14, 4195. ".. The reason why mRNA-LNP vaccines are susceptible to oxidation/hydrolysis is that they are	

	sensitive to temperature changes and contain lipids with unsaturated acyl chains.” (iv) Pharmaceutics 2024, 16, 131. “... there is a noticeable challenge regarding lipid excipient chemistry and its performance, particularly concerning oxidative stability. Antioxidants have been utilized to preserve the stability of pharmaceutical products by intervening chemically in crucial oxidative phases... Optimal antioxidant selection involves assessing solubility within the intended formulation phase, particularly in areas where the oxidation-sensitive substrate (e.g., active pharmaceutical ingredient) is expected to be present.”
--	--

References:

- [1] Moderna Patent No. WO2018089540A1, Figure 4. “Antioxidant comprises ascorbic acid, citric acid, malic acid, methionine, monothioglycerol, phosphoric acid, potassium metabisulfite, alpha-tocopherol, or any combination thereof.”
- [2] CureVac Patent No. WO2011144358A1.
- [3] Moderna Patent No. WO2017218704A1. “Examples of antioxidants include, but are not limited to, alpha tocopherol, ascorbic acid, ascorbyl palmitate, butylated hydroxyanisole, butylated hydroxy toluene, monothioglycerol, potassium metabisulfite, propionic acid, propyl gallate, sodium ascorbate, sodium bisulfite, sodium metabisulfite, and/or sodium sulfite. Examples of chelating agents include ethylenediaminetetraacetic acid (EDTA), citric acid monohydrate, disodium edetate, dipotassium edetate, edetic acid, fumaric acid, malic acid, phosphoric acid, sodium edetate, tartaric acid, and/or trisodium edetate.”
- [4] Moderna Patent No. WO2018170336A1, WO2019046809A1, WO2020061457A1, WO2020061457A1, WO2017049245A2. “... Preservatives may include, but are not limited to, antioxidants, chelating agents, antimicrobial preservatives, antifungal preservatives...”
- [5] Protiva Biotherapeutics Patent No. US20130022649A1, “Snalp formulations containing antioxidants.”
- [6] Alnylam Patent No. US8158601B2, “Lipid formulation.”
- [7] *Journal of Pharmaceutical Sciences* **2024**, 113, 377. “Factors Affecting Stability of RNA – Temperature, Length, Concentration, pH, and Buffering Species.”
- [8] Biontech SE, Patent No. US20240033344A1 “Pharmaceutical compositions comprising particles and mRNA and methods for preparing and storing the same”
- [9] GlaxoSmithKline, Patent No. WO 2023/021427 A1 “Freeze-drying of lipid nanoparticles (LNPs) encapsulating RNA and formulations thereof”
- [10] *Bioscience Reports*, **2020**, 40, BSR20193767. “Comprehensive analysis of PPAR γ agonist activities of stereo-, regio-, and enantio-isomers of hydroxyoctadecadienoic acids.”
- [11] *Prostaglandins, Leukotrienes, and Essential Fatty Acids*, **2009**, 81, 53. “Activation of the antioxidant response element by specific oxidized metabolites of linoleic acid.”
- [12] *Adv. Funct. Mater.* **2022**, 32, 2106727. “Discovery of a Novel Amino Lipid That Improves Lipid Nanoparticle Performance through Specific Interactions with mRNA.”
- [13] *Nature Communications* **2021**, 12, 6777. “A novel mechanism for the loss of mRNA activity in lipid nanoparticle delivery systems.”
- [14] *Biomed Chromatogr.* **2013**, 27, 422. “Identification and profiling of targeted oxidized linoleic acid metabolites in rat plasma by quadrupole time-of-flight mass spectrometry.”

More specific comments:

This is interesting but this phenomenon is specific for unsaturated lipids such as Dlin-MC3-DMA since the alkene groups are prone to oxidize over the time.

We certainly agree with the Reviewer that large aspects of the manuscript are specific to unsaturated lipids; however, degrees of unsaturation are ubiquitous throughout lipid libraries despite their susceptibility towards oxidation due to their propensity to increase membrane disruption. For example, in a recent Comment reviewing ionizable lipids under clinical development for RNA therapeutics, four of the eight lipids contain degrees of unsaturation (*Nat Commun.* **2021**, 12, 7233). Additionally, degrees of unsaturation can be found in helper lipids, for example DOPE. This is discussed within the manuscript, Lines 79–86: “double bonds may make MC3 more susceptible to oxidation... these functional motifs increase nucleic acid delivery efficiency by

enhancing the fusogenicity of the lipid⁸. The kinked molecular shape of the cis double bond being crucial for potency may be a generalizable design rule... As such, strategies to mitigate the instability of ionizable lipids may be preferred over an expectation that problematic functionalities simply not be explored within lipid libraries.”

1. Oxidation of ionizable lipid through amine of the lipid followed by adduct formation has been predicted previously (Ref 2). Did author notice such phenomenon with siRNA and MC3 lipid?

We thank the Reviewer for their question. As the Reviewer mentions, work by Moderna has elucidated RNA-lipid adduct formation through a pathway invoking N-oxidation of the amine head group. Although we noted both the N-oxidized head group of MC3 (*e.g.*, degradant #20 in Supplementary Appendix), as well as a demethylated byproduct of MC3 that could result from downstream degradation (*e.g.*, degradant #21 in Supplementary Appendix), we did not observe the production of an aldehyde-containing MC3 derivative, nor an RNA-lipid adduct with a molecular weight corresponding to reaction with an aldehyde-containing moiety. Thus, our work represents a distinct route of RNA-lipid adduct formation taking place through the reactive lipid tail, as opposed to the amine head group. Distinctions from Moderna's work is detailed in Lines 366–372. All degradants, corresponding molecular weights, and examples of theoretical structures (other skeletal isomers may exist) are now included in the revised manuscript (Supplemental Appendix). Additionally, descriptions within supplementary tables have been updated to aid the reader.

2. Did authors investigate other saturated lipids such as Pfizer/Moderna COVID-19 lipids and siRNA? Can the proposed mechanism work with histidine buffer?

We thank the Reviewer for this experimental suggestion. To address the Reviewer's question, we formulated mRNA-LNPs with the Pfizer (ALC-0315) and Moderna (SM-102) lipids and directly compared to LNPs formulated with MC3 in the two relevant buffers: PBS and histidine buffer (see Supplementary Figure 15, Supplementary Table 5 below). Resulting mRNA-LNPs across the three ionizable lipids all had similar initial sizes between 65–85 nm, PDIs below 0.3, and encapsulation efficiencies above 80%. Particles were stored at 5 and 25 °C and critical quality attributes were analyzed after two and four weeks. Only mRNA-LNPs formulated in PBS with MC3 lipid demonstrated instability at 25 °C (*e.g.*, particle size increased and encapsulation efficiency decreased), while those stabilized by either ALC-0315 or SM-102 remained stable. However, all formulations in histidine buffer were stable. This data is now provided within Supplementary Figure 15. Additionally, integrity of each ionizable lipid was evaluated and is now included in Supplementary Table 5. Again, only MC3-stabilized mRNA-LNPs in PBS demonstrated meaningful degradation over time (*i.e.*, falling from 94% after formulation to 58% after 4 weeks at 25 °C). These data are consistent with the first point the Reviewer makes—namely, that the optimized drug product matrix and use of antioxidants is largely relevant for lipids with unsaturated bonds. This is now referred to within the conclusion of the main text, namely Lines 444–448:

Preliminary work demonstrates that these formulation solutions can be applied more sensitive cargoes, including mRNA, though benefits are dependent on ionizable lipid chemistry and are most pronounced with unsaturated lipids

(Supplementary Figure 15, Supplementary Table 5). More comprehensive evaluations of these systems are underway, which are likely to represent new stability challenges inherent to the drug substance...

Supplementary Figure 15. Stability of EPO mRNA-LNPs formulated in phosphate- or histidine-containing buffer using three different ionizable lipids: MC3 1, ALC-0315 or SM-102. Particle size of mRNA-LNPs was evaluated at (A) 5 °C or (B) 25 °C. Encapsulation efficiency of mRNA-LNPs was evaluated at (C) 5 °C or (D) 25 °C. Legend: initial time (black bars), 2 weeks at storage temperature (orange bars), 4 weeks at storage temperature (red bars). Lipid integrity of each formulation is included within Supplementary Table 5.

Supplementary Table 5. Comparing the stability of three different ionizable lipids (MC3 (1), ALC-0315, and SM-102) stabilizing EPO mRNA-LNPs within phosphate and histidine-containing buffers. RNA-LNP solutions were stored at either 2–8 °C or room temperature (22–25 °C).

Intact Ionizable Lipid	EPO mRNA-LNPs in PBS 1X, pH 7.4					EPO mRNA-LNPs in 10 mM Histidine + 140 mM NaCl, pH 6				
	T0	2–8 °C, 2w	2–8 °C, 4w	RT, 2w	RT, 4w	T0	2–8 °C, 2w	2–8 °C, 4w	RT, 2w	RT, 4w
MC3 (1) Intact	93.91	91.54	88.87	62.13	57.78	95.74	95.72	95.66	95.82	95.86
ALC-0315 Intact	99.94	99.93	99.93	99.94	99.94	99.9	99.91	99.93	99.92	99.9

SM-102 Intact	99.68	99.68	99.61	99.62	99.76	99.68	99.75	99.69	99.51	99.69
------------------	-------	-------	-------	-------	-------	-------	-------	-------	-------	-------

3. Figure 5, siRNA functionality, did authors evaluate the activity of LNPs in other buffers reported in table 1?

We did not evaluate the activity of LNPs in other buffers reported within Table 1, as these buffers were not our lead formulation matrix. If the Reviewer believes this data would be significant value, we could generate these data.

4. Figure S13, why one lipid tail involved in adduct formation with siRNA?

We thank the Reviewer for raising this question. We reason that this is because degradants containing one oxidation event at a single tail are much more abundant than degradants with two oxidation events across both tails (corroborated by Supplementary Table 3 and Supplementary Appendix). However, we have revised Supplementary Figure 13, as well as Supplementary Table 2, to clarify that lipidation of RNA can occur more than once. This is now explicitly called out in the main text, Lines 355–358:

Lipidation of RNA occurred up to two times per strand, albeit less commonly (e.g., 13.2% and 2.4% of antisense strand was lipidated once or twice, respectively, after 6 months of storage in PBS 1X (Supplementary Figure 13, Supplementary Table 2)).

In summary, the manuscript should be rejected for publication in Nature Communications. It may be suitable for a more specialized journal, as a kind of review supported by the group own data.

Reviewer #2 (Remarks to the Author):

The manuscript by Estabrook et al. focuses on demonstrating that lipid nanoparticle (LNP) formulations containing MC3 lipids could react with RNA cargoes. The suggested mechanism is the oxidation of MC3 lipid tails resulting in production of a dienone species and generating an electrophilic degradant that reacts with RNA to form lipid-RNA adduct. The authors also suggest modifications to the LNP formulation that may remedy this problem.

There is much interest in both making the LNP formulations more efficient and shed light on the mechanisms behind instability and inefficiencies of the existing formulations and thus this manuscript deals with a significant issue. On the other hand, there are major issues with the manuscript that need to be addressed. I will focus on the computational side of the work given my own expertise.

The main issue with the computational side of this work is the lack of clarity of the methodology. There is very little information provided on the details of both the DFT and MD methods. For instance, it is not even stated how long are the MD simulations and how many times they have been repeated and how the systems were set up. Without

knowing the basic details of the simulations and analysis techniques it is not possible to determine whether the conclusions are sound or not.

We thank the Reviewer and regret that the original version of the manuscript lacked critical information on the DFT and MD methods. Accordingly, further information has now been included within the main manuscript under the “Computational details and methodology” section. For DFT, we have provided more details on how we calculated bond dissociation energies and free energies along with additional references. For MD, our revised version provides a detailed explanation of how we set up the simulation systems and perform MD simulations. To name a few, we employ CHARMM36 force field and its extended version for the modified RNA (*J. Comput. Chem.* **37**, 896 (2016)) to model siRNA, and CHARMM General Force Field is used to parameterize the cationic ionizable lipids. We hope that this additional information will allow the Reviewer to accurately assess the validity of the conclusions reached.

Another issue is the use of CGenFF for the lipids and RNA forcefield. It is strange that the authors have used a generic force field for the most important part of the simulation while there are more specific forcefields for lipids and RNA.

We appreciate the reviewer for raising this concern, and we have revised our manuscript to clarify the parameters used. We used Charmm36 and its extension for modified nucleotides to model our siRNA, and we applied CGenFF parameters for the lipids. We are aware of concerns regarding the accuracy of Charmm for RNA compared to Amber, particularly the underestimated interactions between base pairs that can lead to unintended annealing (*Chem. Rev.* **118**, 4177 (2018)). However, our siRNA maintained its double helix form throughout the entire simulation, so we believe the accuracy of the Charmm parameters for nucleotides is sufficient for our study. As there are no specialized CHARMM parameters for MC3 and the MC3 byproduct, we employed CGenFF to parameterize the lipids. Dr. Im’s group at Lehigh University, a core member of the CHARMM family, also used CGenFF for modeling MC3 lipids in their bilayer simulations (*J. Chem. Inf. Model.* **61**, 5192 (2021)). Therefore, we believe that applying CGenFF for lipids alongside the regular CHARMM force field for RNA is a reasonable choice; however, we agree with the reviewer that the choice of force field could potentially lead to different results. If the reviewer suggests it, we are willing to perform a force field comparison study by conducting a separate set of MD simulations using the OPLS force field included in the Schrödinger suite, which is another force field for which Lilly holds the license; however, we did not believe a force field comparison to be necessary to support our conclusions.

Based on these observations the computational part of this manuscript is not acceptable in my opinion.

Reviewer #3 (Remarks to the Author):

Reviewer Comments

Thank you to the authors for sharing this interesting work and for investigating how oxidation of unsaturated ionizable lipids can contribute to the loss of RNA LNP potency. In general, the study is interesting and the chemical analysis was interesting. However, there are no statistical analyses in this study; these tests are needed to help support some of the major interpretations of key data. Additionally, a few controls should be added. Please consider addressing the comments below to improve this manuscript. **We are sincerely grateful for the level of attention that this Reviewer has given the manuscript and are glad that they found the findings of interest to them. Accordingly, we have done our best to address each of the suggested revisions and additional experiments.**

Line 133: beyond citing it is the likely buffer of Onpattro, please comment on why the calcium and magnesium would be omitted.

As the reviewer mentions, calcium and magnesium were primarily omitted to mimic the formulation of Onpattro, which does not contain these two metals, and these two inorganic metals are not common excipients within the field. Additionally, we were concerned that (i) calcium phosphate aggregates could obfuscate the long-term stability data, and (ii) these metals could chelate with other additives explored within Table 1 (e.g., citrate and Tris). We did not expect either calcium or magnesium to provide a significant benefit and thus chose not to include them.

Line 146-158: the authors say that nanoparticles were aggregating and perform measurements by MFI to show an increase in the number of particles that are $\geq 1\mu\text{m}$. The authors said that sub 100 nm particles were aggregating. In its current form, I don't think the presented data supports this claim. The DLS data does show a decline in intensity over time of the nanometer-scale nanoparticles, but intensity values aren't necessarily comparable between conditions. If the DLS used a different level of attenuation, would this make intensity numbers incomparable? I suggest the authors measure nanoparticle distribution by an orthogonal method, such as the NanoSight machine and **check for a decline in the sub 100 nm population**. Once done, please revise these statements accordingly.

In retrospect, we agree with the Reviewer that we were too confident in the initial assessment of particle aggregation and thank the Reviewer for pointing this out. The Reviewer is also correct that, unfortunately, the DLS used a different level of attenuation throughout the month-long studies and thus interpreting the intensity data was non-trivial. We have taken the Reviewer's suggestion and reprepared siHPRT-LNPs in PBS 1X, then performed an accelerated stress test by holding them at 40 °C over 7 days, with storage at 5 °C storage as a control. Nanoparticle tracking analysis via NanoSight is now included as Supplementary Figure 4:

Supplementary Figure 4. Nanoparticle tracking analysis of siHPRT-LNPs in PBS 1X for 1 week at either 5 or 40 °C. Aliquots for each replicate and temperature are derived from the same master stock solution. (A) Particle concentration by size for LNPs held at 5 °C (blue line) or 40 °C (red line). No meaningful count of particles between 300–1000 nm was detected (data omitted). (B) Particle concentration (<1000 nm) for siHPRT-LNPs held at 5 and 40 °C. Data are an average of 10 runs between two experimental duplicates, with error bars representing the standard deviation. Significance is determined by a two-tailed Student's t-test of unequal variance, $p \leq 0.001$ ***.

We believe that these results corroborate our initial hypothesis and thank the Reviewer for the recommendation of evaluating particle stability through an orthogonal method. The discussion within the main text has been updated accordingly:

“Nanoparticle tracking analysis of RNA-LNPs held at either 5 or 40 °C over one week demonstrated a statistically significant decrease in sub-micron particles, particularly those around ~70 nm, for samples at elevated temperatures (Supplementary Figure 4). Collectively, these data corroborate that smaller particles gradually aggregate at higher temperatures to form larger, micron-sized aggregates in phosphate-based buffers.”

Line 159 – 162 and Figure 2: please perform the appropriate statistical tests to show that the EE and RNA content remained the same.

To address the Reviewer's concerns, we evaluated critical quality attributes against storage time for each temperature. Additionally, phrases like “unchanged” were softened to “were similar.” Each data set was evaluated by linear regression analysis and ANOVA p-values are now reported in the revised text, as highlighted below:

“RiboGreen assays were performed and revealed that encapsulation efficiencies (EE%) were similar over time regardless of storage temperature (Figure 2E, 5 and 25 °C had p-values of 0.12 and 0.45, respectively). By the same analysis, the RNA content remained similar in all samples over time (Supplementary Figure 4, 5 and 25 °C had p-values of 0.38 and 0.85, respectively).”

Line 191-193: can the authors please show the LC/MS of their starting MC3 materials (MC3 in solution and not formulated into an LNP), to determine if it arrived oxidized or hydrolyzed? Also, separately oxidize and hydrolyze samples of MC3 and show the LC/MS. These data should be included as positive controls.

We thank the Reviewer for their suggestions and have performed the recommended experiments. The revised Supplementary Figure 6 now includes LC/MS analysis of MC3 in solution, including upon arrival (initial time T0), and after thermal stress in solution (2–8 °C, 25 °C, and 40 °C over 4 weeks). Generally, upon arrival we observe that the primary degradant byproducts of MC3 are oxidized degradants and cumulatively represent ~3.5% of observed species, while no significant hydrolytic byproducts were observed. Extracted ion chromatograms from the LC/MS are now also included in Supplementary Figure 6 for representative samples, including the starting material (MC3, T0). Additionally, per the Reviewer's request we have stressed MC3 in solution under varying forced degradation conditions. These data are now included in new Supplementary Figure 14. Here, ethanolic MC3 was solubilized in ethanol and diluted in either PBS or the lead histidine formulation. The solutions were then exposed to (i) metals (1 ppm Ni, Fe and Cu), (ii) peroxides (1 ppm H₂O₂), or pH-adjusted (iii) acidic (pH 5) and (iv) basic (pH 8) formulations. After 2 weeks, degradation across temperatures was analyzed. Examples of LC/MS extracted ion chromatograms are included within Supplementary Figure 14 B–D. Generally, we observed that MC3 was particularly sensitive to metals and acidic pH, with histidine effectively protecting against the former, presumably due to its ability to act as a chelator (see updated discussion and SI Figure callouts within the main text):

“Aside from histidine, the excipients that promoted stability best included methionine, tryptophan, EDTA, and citrate. We believe that the latter two excipients impart particle stability by chelating trace metals that are otherwise capable of catalyzing lipid oxidation, as forced degradation experiments on MC3 1 in solution (not formulated in an LNP) demonstrated susceptibility to oxidation in solutions containing 1 ppm nickel, iron and copper (Supplementary Figure 14).”

Degradants promoted by metal inclusion were primarily oxidized degradants (mono oxidation at the tail, with the most abundant being 2, mono oxidation at the head, double and triple oxidized variants), though tail cleavage was also observed. These results reinforced the results found in Supplementary Figure 6 and the results for MC3 integrity after formulation in LNPs (Supplementary Table 1), namely that hydrolytic species are consistently rare in comparison to oxidized degradants.

Line 212-213: please clarify or better support the supposition that oxidation likely happens within the LNP core. Is the idea that the core contains MC3 that would react

with the RNA? Its ok to be speculative, but please describe this spatial hypothesis in better detail.

Here, we meant that because (i) we know the ionizable lipids and RNAs are oxidized in these LNP systems, and (ii) per literature precedent, ionizable lipids and RNAs are localized in the core of LNPs, we thus speculate that oxidation events likely also take place in the core (as opposed to in bulk solution). Although the proximity of ionizable lipids and RNA cargoes to one another may be critical (e.g., a catalytic mechanism whereby ionizable lipid oxidation generates reactive oxygen species that then react with nearby RNA), here we simply meant that oxidation is likely taking place where the molecules most likely reside. We have updated the main text to clarify our statements and correctly cite the works that corroborate this speculation:

Granted that these RNA-LNPs have a high encapsulation efficiency and there is precedent suggesting ionizable lipids and analogous RNA cargoes tend to organize within the LNP core²³⁻²⁵, we speculate that the oxidation of these two molecules also occurs within the core (and not, say, within bulk solution).

Line 214-216: These statements on concentration-dependent oxidation don't appear to be supported by statistical analysis of the Figure S6 data. Please increase the number of replicates for S6 and perform the need statistical analysis. Alternatively, reconsider the inclusion of these statements. Perhaps add in an oxidizing agent to create a positive control for oxidation, which would benchmark the loss of potency (change in IC50) that is caused by oxidation.

We thank the Reviewer for the suggestion. Due to the 6-month time course of the mentioned supplemental Figure and limited overall value to the manuscript, we have chosen to retract these statements.

Lines 230-233 and Figure 3: no statistical analysis was done to support the statement that sizes and PDI were unchanged. Please address this issue.

We have now included p-values where necessary through a two-tailed Student's t-test assuming an unequal sample variance. The main text has been revised:

At the initial time point, RNA-LNPs in histidine-containing buffers were visually comparable to those formulated in PBS (Figure 3A), had similar sizes (Figure 3B,C, p-value of 0.96) and more narrow polydispersities (Figure 3D, p-value of 0.02), as corroborated by cryoEM analysis.

Line 383-385: Figure 5B shows error bars, but no statistical analysis is done. Also, the statement says that room temperature LNPs in histidine buffer showed no loss in potency; however, the Figure 5B table shows everyone to be within 0.01 of each other. Please reconsider these statements after statistical analysis.

We have revised the main text discussion. To clarify, the fact that the IC₅₀ of LNPs after storage at room temperature in histidine buffer (0.06 nM) is similar to the IC₅₀ of fresh LNPs (0.05 nM) is supportive of the notion that potency is retained. By comparison, an IC₅₀ value for LNPs at room temperature in PBS cannot be extrapolated because of the severe loss in potency. This is now further supported by statistical analysis of relative HPRT expression at a given concentration, namely 0.1 nM:

On the other hand, LNPs stored in histidine buffer at room temperature for one month (red line) remarkably retained potency, with IC₅₀ values similar to those of fresh or refrigerated LNPs (Figure 5B). For example, at a concentration of 0.1 nM siRNA, HPRT expression is effectively knocked down to 42 ± 2% for fresh LNPs, LNPs refrigerated in PBS (ns, p-value of 0.65), and LNPs refrigerated or at room temperature in histidine buffer (ns, p-values of 0.85 and 0.79). Conversely, HPRT expression after incubation with LNPs at room temperature in PBS has negligible knockdown, with a relative HPRT expression of 96% (p-value of 0.02 when compared to fresh LNPs). These data support that RNA-LNPs stored in histidine-containing buffers retain their biological function even at elevated temperatures, while those stored in phosphate-containing buffers undergo a temperature-dependent loss in potency.

Figure S2: please clarify if this was N=1, or an average of a triplicate measurements, etc.

For all following comments, we regret the lack of clarity in the original version of the Supporting Information and appreciate the Reviewer's diligence. These data are an average of four independent samples, and the Figure caption has been updated accordingly.

Figure S6: Indicate the number of replicates.

These data are an average of three independent samples, and the Figure caption has been updated accordingly.

Figure S9B: difficult to see axis labels. When one zooms in, it is fuzzy and one cannot make out the numbers.

A higher resolution image of Supplementary Figure 9B is now included such that chemical shift assignments should be legible.

Figure S12A: Thank you for providing the sequence. However, please indicate what upper and lower case letters mean for the siRNA, and what the modification abbreviations mean.

A legend has been added to Supplementary Figure 12, namely:

Legend: Upper case letters indicate identity of the nucleoside (U: uracil; C: cytosine; A: adenine; G: guanine). Lower case letters in front of the nucleoside represent modifications to that nucleoside at the 2' position (mX: 2'-O-methyl ribonucleoside; fX: 2'-fluoro-deoxyribonucleoside). Lower case letters after the

nucleoside represent modifications to the phosphate linkage (Xs: phosphorothioate linkage).

Figure S13: the LC/MS traces are very low resolution and difficult to appreciate; they become fuzzy when one zooms in.

Higher resolution LC/MS traces are now included within Supplementary Figure 13 (and throughout the supporting information) such that all masses should be legible.

Tables S1 to S4. The nomenclature and shorthand used in the leftmost column and header rows aren't fully explained. Either rename them or label them more clearly. For example, in the Table S4 "siRNA Degradation" column, what does each row entry mean?

We thank the Reviewer for the feedback. For Tables including LC/MS analysis of lipid degradants (Supplementary Table 1 and Supplementary Table 3), the shorthands have been replaced with more descriptive labels. Additionally, examples of structures that match massed identified through LC/MS analysis of DLin-MC3-DMA (1) are included in the Supplementary Appendix. For Tables including LC/MS analysis of RNA and RNA-lipid degradants (Supplementary Table 2 and Supplementary Table 4), a detailed legend is now included.

“The Impact of Buffer in RNA-LNP Formulations: Mitigating Oxidation of Ionizable Lipid Tails and RNA-Lipid Adduct Formation”.

Point-by-point responses by the authors (in red) to the reviewers' comments, reproduced verbatim.

Reviewer #2 (Remarks to the Author):

The authors have addressed my concerns.

Reviewer #3 (Remarks to the Author):

Thanks for working to extend the study and address all previous reviewer comments. I am unable to comment on the simulation work, so focused on comments from Rev 1 and Rev 3.

For Rev 3, given the retraction and attenuation of the conclusions, this is probably not suitable for Nature Communications.

- the concentration-dependent data was compelling, but now it has been retracted due to experimental limitations

We apologize for this confusion. These data were retracted due to our interpretation of Reviewer 3's previous comment to perform statistical analysis or alternatively "reconsider the inclusion of these statements." Unfortunately, each data point was an individual sample. As statistical analysis was not possible and we could not repeat the study, we interpreted that Reviewer 3 would rather us retract the Figure. However, if the Reviewer finds it valuable, we can include it again. In the authors' opinion, while the concentration-dependent degradation data was interesting, it added only modest value to the overall manuscript.

- the authors claim that smaller ~70 nm particles are aggregating over time, causing a loss in this population. however, we cannot see an increase in larger particles, and the authors say no meaningful count of particles between 300–1000.

To clarify, a meaningful increase in larger particles is observed (A) qualitatively by DLS from 100–1000 nm (Supplementary Figure 2) and (B) quantitatively by MFI for particles >2 μm (Supplementary Figure 3); however, the Reviewer is correct that we acknowledged no meaningful count in 300–1000 nm particles being observed by NTA (Supplementary Figure 4). We attribute this apparent inconsistency to the lower particle quantitation limit of NTA. For example, the difference in particle concentration between samples stored at 1 week at 5 °C and 1 week at 40 °C is approx. $2.5\text{E}8$ particles/mL. If we assume that the decrease is solely due to 70 nm particles aggregating to form 300 nm particles, then we would expect to see an increase in the number of 300 nm particles of approximately $3.2\text{E}6$ particles/mL which is close to the lower particle concentration specification ($1\text{E}6$ particles/mL) for the instrument. For 70 nm particle aggregating to 1000 nm particles, the increase would be approx. $8.6\text{E}4$ particles/mL

which is well below the quantitation limit of the instrument. Thus, to address the Reviewer's previous concerns we have leveraged data from complementary techniques (DLS, MFI and NTA) to capture particle stability across a wide range of sizes. We do believe the conclusions stated within the main text ("smaller particles gradually aggregate at higher temperatures to form larger, micron-sized aggregates in phosphate-based buffers") are justified by the current data set. In the authors opinion, softening the language to allow for alternative methods of particle size increases (like Ostwald ripening, rather than aggregation) would not diminish the impact of the manuscript, which does not lie in detailing the exact mechanism of nanoparticle destabilization. Thus, if the Reviewer agrees then the language could be softened.

- Thanks for incorporating statistical analysis of Figure 2 and other figures. However, the figure captions must be updated to include statistical analysis info (not just the main text).

We apologize for this oversight. The relevant figure captions have been updated to include information on statistical analysis (Figure 2, Supplementary Figure 5, Figure 3, and Figure 5).

Seeing the feedback from Rev 1 and the rebuttal, the work should probably test more unsaturated lipids to show this phenomena is broadly applicable, and the degree to which it is a problem.

- There are no methods around the production of the SM-102 and ALC-0315 LNPs. Please ensure the sequences of all RNA used, including mRNA, are reported.

We regret the omission of these details and have now included them within the Materials and Methods. The SM-102 and ALC-0315 particle production is captured under the RNA-lipid nanoparticle formation procedure, while the mRNA sequence information is captured within Materials and equipment. Both are included below for ease of reference. Please note that the full sequence of commercial, off-the-shelf EPO mRNA is proprietary to TriLink Biotechnologies, so we are unable to entirely comply with that request:

Methods of production of SM-102 and ALC-0315 lipid nanoparticles: "For ALC-0315 nanoparticles, the above procedure was followed with ALC-0315 (Echelon Biosciences) in place of D-Lin-MC3-DMA, and ALC-0159 (Echelon Biosciences) in place of DSPC. mEPO (CleanCap) was utilized as the cargo, with an N:P ratio of 6.0. mEPO working solution was prepared by diluting 593 μ L of 1 mg/mL stock to 9 mL for a working concentration of 0.0659 mg/mL. Lipids were combined for a total lipid concentration of 8 mM, with a molar ratio of 46.3:42.7:9.4:1.6 (ALC-0315:Cholesterol:ALC-0159:DMG-PEG-2000). Total batch size was 12 mL, collected in a 15 mL Falcon tube. SM-102 nanoparticles were formulated using the same procedure, with SM-102 (Echelon Biosciences) in place of ALC-0315, and DSPC in place of ALC-0159. Additionally, SM-102 nanoparticles were formulated with a molar ratio of 50:38.5:10:1.5 (SM-102:Cholesterol:DSPC:DMG-PEG-2000)."

Sequence of mRNA: “EPO mRNA (CleanCap EPO, SKU L-7209, Lot# WOTL76618) was purchased from TriLink Biotechnologies and has full length of 859 nucleotides with a proprietary sequence, and an open reading frame length of 582 nucleotides with the following sequence:

```
AUGGGCGUGCACGAGUGCCCCGCCUGGCUGUGGCUGCUGCUGAGCCUGCUGA
GCCUGCCCCUGGGCCUGCCCGUGCUGGGCGCCCCCCCCCGGCUGAUCUGCGA
CAGCCGGGUGCUGGAGCGGUACCUGCUGGAGGCCAAGGAGGCCGAGAACAUCA
CCACCGGCUGCGCCGAGCACUGCAGCCUGAACGAGAACAUCACCGUGCCCGAC
ACCAAGGUGAACUUCUACGCCUGGAAGCGGAUGGAGGUGGGCCAGCAGGCCGU
GGAGGUGUGGCAGGGCCUGGCCUGCUGAGCGAGGCCGUGCUGCGGGGCCAG
GCCUGCUGGUGAACAGCAGCCAGCCUGGGAGCCCCUGCAGCUGCACGUGGA
CAAGGCCGUGAGCGGCCUGCGGAGCCUGACCACCCUGCUGCGGGGCCUGGGC
GCCAGAAGGAGGCCAUCAGCCCCCGACGCCGCCAGCGCCGCCCCCCUGCG
GACCAUCACCGCCGACACCUUCCGGAAGCUGUCCGGGUGUACAGCAACUCC
UGCGGGGCAAGCUGAAGCUGUACACCGGCGAGGCCUGCCGGACCGGCGACCG
GUGA.”
```

- As a control, testing unsaturated lipids make sense. But to increase impact, showing that this mechanism applies to all unsaturated lipids would be of interest; otherwise, the impact feels limited the data generated by MC3 only.

The big idea is that unsaturated tails are an issue, but without verifying this in other configurations (e.g. would an unsaturated bond that is sterically hindered cause the same issues, or must it be in a chain that is accessible?). The work's impact would increase if verified across other (FDA approved or not) unsaturated lipids. The authors mentioned that new libraries have unsaturated lipids in them. Trying an extra two from these libraries could be great.

We greatly appreciate the Reviewer's experimental suggestion to try additional lipids with unsaturated bonds and for the Reviewer's patience as we executed these studies. Towards these ends, we sourced three cationic/ionizable lipids: DOTAP, DODMA, and KC2-DLin-DMA. These lipids systematically vary in degrees of unsaturation within the lipid tail (alkene vs. 1,4-pentadiene) and linker chemistries (ester, ether, and cyclic acetal). We formulated these lipids into siRNA-LNPs, buffer exchanged the product into either phosphate- or histidine-containing buffers and evaluated product stability over four weeks. To accelerate degradation and explore lipid sensitivities to environmental stressors, samples were stored at elevated temperatures of 25 °C and 40 °C and select samples were spiked with either 1 ppm hydrogen peroxide or metals (nickel, iron and copper, 1 ppm each). To ensure this study was robust, we evaluated stability through the following readouts:

- (i) colloidal stability (particle size and polydispersity) by DLS,
- (ii) RNA payload integrity (sense and antisense strand integrity) by a RP-IP-HPLC denaturing method, and
- (iii) cationic/ionizable lipid integrity through LC-MS.

These data are now included through addition of Supplementary Figures 9–11 and Supplementary Tables 3–5. In brief, each of these three distinct RNA-LNPs demonstrated enhanced stability in histidine-containing buffers when compared to phosphate. In choosing lipids with different chemistries, we demonstrate that while destabilization mechanisms may vary—for example, in the number of alkenes within the lipid tail prone to oxidation, or the susceptibility of the linker to hydrolysis (esters versus ether versus cyclic acetal linkers)—all formulations benefit from histidine-containing buffers. We believe the inclusion of these experiments further supports the generalizability of histidine-containing buffers across this class of RNA-LNPs and thus increases the impact, as the Reviewer had suggested. A discussion is now included in the main text (copied below) and the Materials and Methods section have been updated accordingly:

To demonstrate the generalizability of the histidine buffer platform, we systematically explored three additional cationic/ionizable lipids: DOTAP, DODMA, and DLin-KC2-DMA. We formulated siHPRT-LNPs with each lipid and buffer-exchanged the resulting particles into either PBS or the histidine-NaCl buffer. To accelerate degradation, select samples were spiked with 1 ppm of either hydrogen peroxide or metals (nickel, iron and copper). Particles were then stored at either 25 or 40 °C for up to four weeks, and analyzed for changes in colloidal stability, RNA payload integrity, and ionizable lipid integrity. Starting with DOTAP, we observed changes in RNA-LNP stability for particles stored in PBS, including particle size, payload and ionizable lipid integrity (Supplementary Figure 9). In all instances, degradation could be mitigated by using histidine buffer. Notably, DOTAP lipid was more sensitive to hydrolysis than oxidation, which we hypothesized was due to there being a single alkene in each lipid tail and two ester linkages (Supplementary Table 3). To confirm this, we next leveraged a DODMA ionizable lipid that contains a single alkene in each lipid tail but lacks esters (Supplementary Figure 10). Accordingly, DODMA-LNPs proved more resilient to colloidal instability and ionizable lipid degradation (Supplementary Table 4). However, payload integrity continued to prove problematic for particles stored in PBS spiked with either metals or peroxides, whereas those stored in histidine retained payload integrity across all conditions. Finally, DLin-KC2-DMA, a lipid which preceded the optimization of MC31, was employed due to its lack of esters (instead containing a cyclic acetal linker) and structurally similar bis-allylic dilinoleyl tail (Supplementary Figure 11). DLin-KC2-DMA LNPs demonstrated dramatic colloidal and chemical degradation when stored in PBS, whereas histidine buffers again mitigated these changes over time (Supplementary Table 5); however, histidine buffers were unable to rescue samples spiked with metals. The increased sensitivity of MC3 and DLin-KC2-DMA to oxidation in comparison to DOTAP and DODMA is supported by the fact that lipids with higher degrees of unsaturation oxidize more rapidly due to the weakness of a bisallylic C–H bond compared to an allylic C–H bond.³¹ For example, linoleic acid is known to oxidize faster than oleic acid.³² However, while each cationic/ionizable lipid may differ in its particular degradation

mechanisms, the ability of histidine buffers to improve product stability is shown to be generalizable across this class of LNPs.

A. siHPRT-Lipid Nanoparticles Generated with DOTAP lipid

B. Particle Size

C. Particle Polydispersity Index

D. siHPRT Payload Integrity

E. siHPRT Concentration after 4W

F. Ionizable Lipid Integrity

Supplementary Figure 9. (A) Analysis of DOTAP-stabilized siRNA-LNPs after four weeks across two temperatures and stressed conditions in two storage buffers: (i) PBS 1X pH 7.4 and (ii) 10 mM histidine + 140 mM NaCl, pH 6.0. Colloidal stability was analyzed via (B) particle size analysis and (C) polydispersity index by DLS. DOTAP LNPs demonstrate particle size increases in PBS at 40 °C but retain sizes below <100 nm in histidine buffer. Payload stability was analyzed via RP-IP-UPLC and is demonstrated via (D) chromatograms of both strands and (E) corresponding RNA content. Payload integrity is degraded by ~60% when particles are stored in PBS with

peroxides regardless of temperature, while particles in histidine buffer retain payload integrity >90% under the same conditions. (F) Ionizable lipid integrity analyzed via LC-MS. The amount of intact DOTAP lipid is degraded by ~40% when stored in PBS at 40 °C but integrity is retained >85% in histidine buffer under the same conditions. Note that nearly all of the DOTAP degradation is attributed to hydrolysis (e.g., PBS control at 40 °C is 41.1% degraded, with 40.8% corresponding to hydrolytic cleavage products and 0.3% corresponding to oxidative degradants). The identity and relative amounts of each DOTAP degradant is provided within Supplementary Table 3. See Materials and Methods for detailed information on RNA-LNP formation procedures, DLS, LC-MS, and HPLC analysis.

A. siHPRT-Lipid Nanoparticles Generated with DODMA lipid

B. Particle Size

C. Particle Polydispersity Index

D. siHPRT Payload Integrity

E. siHPRT Concentration after 4W

F. Ionizable Lipid Integrity

Supplementary Figure 10. (A) Analysis of DODMA-stabilized siRNA-LNPs after four weeks across two temperatures and stressed conditions in two storage buffers: (i) PBS 1X pH 7.4 and (ii) 10 mM histidine + 140 mM NaCl, pH 6.0. Colloidal stability was analyzed via (B) particle size analysis and (C) polydispersity index (PDI) by DLS. DODMA LNPs retain particles sizes below <100 nm in both PBS and histidine buffer but PDI is increased slightly to >0.3 when stored in PBS + peroxides. Payload stability was analyzed via RP-IP-UPLC and is demonstrated via (D) chromatograms of both strands and (E) corresponding RNA content. Payload integrity is degraded by ~40% when particles are stored in PBS with metals or peroxides at 40 °C, while particles in histidine buffer retain payload integrity >90% under the same conditions. (F) Ionizable lipid integrity analyzed via LC-MS. The amount of intact DODMA lipid retained >85% in all

buffers and conditions. The identity and relative amounts of each DODMA degradant is provided within Supplementary Table 4. See Materials and Methods for detailed information on RNA-LNP formation procedures, DLS, LC-MS, and HPLC analysis.

A. siHPRT-Lipid Nanoparticles Generated with KC2-DLin-DMA lipid

B. Particle Size

C. Particle Polydispersity Index

D. siHPRT Payload Integrity

E. siHPRT Concentration after 4W

F. Ionizable Lipid Integrity

Supplementary Figure 11. (A) Analysis of DLin-KC2-DMA-stabilized siRNA-LNPs after four weeks across two temperatures and stressed conditions in two storage buffers: (i) PBS 1X pH 7.4 and (ii) 10 mM histidine + 140 mM NaCl, pH 6.0. Colloidal stability was analyzed via (B) particle size analysis and (C) polydispersity index by DLS. DLin-KC2-DMA LNPs demonstrate large particle size and polydispersity increases in PBS at either 25 or 40 °C. Storage in histidine buffer mitigates these size changes and retains particle sizes <75 nm in all conditions except for +metals at 40 °C for 4 weeks. Payload stability was analyzed via RP-IP-UPLC and is demonstrated via (D) chromatograms of both strands and (E) corresponding RNA content. With or without peroxide spiking, payload integrity falls below 30% when particles are stored in PBS regardless of temperature or stress, while particles in histidine buffer retain payload integrity >87% under the same

conditions. However, histidine does not demonstrate sufficient payload protection against the severe degradation that metals induce. (F) Ionizable lipid integrity analyzed via LC-MS for the major DLin-KC2-DMA variant observed (-28 Da compared to the expected MW implies two methylenes shorter than advertised). With or without peroxide spiking, the amount of intact lipid falls to ~50–64% in PBS; conversely, ionizable lipid integrity is retained >85% in histidine buffer under the same conditions. However, histidine buffer cannot efficiently protect against metal spiking, with integrity <35% regardless of buffer. The identity and relative amounts of each DLin-KC2-DMA degradant is provided within Supplementary Table 5. See Materials and Methods for detailed information on RNA-LNP formation procedures, DLS, LC-MS, and HPLC analysis.

Supplementary Table 3. LC/MS analysis of DOTAP lipid and degradants after siHPRT-LNP storage over four weeks at 25 or 40 °C in two different buffers: (i) phosphate-buffered saline (PBS) 1X, pH 7.4 (“PBS”) or (ii) histidine (10 mM), pH 6.0 (“His”). Selected data from this table is presented in Supplementary Figure 9.

DOTAP	Initial Time (T0)		4W at 25 °C						4W at 40 °C					
	PBS	His	PBS	PBS	PBS	His	His	His	PBS	PBS	PBS	His	His	His
Degradations	Control	Control	Control	Metals	Peroxides	Control	Metals	Peroxides	Control	Metals	Peroxides	Control	Metals	Peroxides
Head Oxidation	0	0	0.03	0.07	0.1	0.02	0.01	0.01	0.18	1.28	0.28	0.05	0.07	0.08
Loss of Oleate + Dienone	0	0	0.04	0.08	0.36	0	0	0	0.1	0.05	0.39	0.02	0.01	0.01
Loss of Oleate + Mono Oxidation	0	0	0.01	0.02	0.08	0	0	0	0.04	0.04	0.15	0.03	0.01	0.01
Loss of Oleate	1.34	1.07	16.59	16.84	16.96	2.86	2.83	5.92	40.58	38.25	41.77	9.12	12.49	14.66
Double Oxidation	0.01	0	0	0.01	0.01	0	0.01	0.02	0.01	0	0	0.01	0.01	0.02
Mono Oxidation at Tail (Dienone)	0.1	0.05	0.1	0.09	0.26	0.05	0.06	0.15	0.08	0.13	0.21	0.06	0.16	0.17
Mono Oxidation at Tail (+O)	0.01	0.01	0.02	0.02	0.03	0.04	0.04	0.11	0.01	0.02	0.02	0.04	0.16	0.16
Double Oxidation at Tail (+2O)	0	0	0.04	0.01	0.15	0	0.01	0.01	0.06	0.05	0.06	0.02	0.04	0.03
Intact	98.54	98.86	83.16	82.86	82.05	97.01	97.04	93.8	58.93	60.18	57.12	90.65	87.05	84.87

Supplementary Table 4. LC/MS analysis of DODMA lipid and degradants after siHPRT-LNP storage over four weeks at 25 or 40 °C in two different buffers: (i) phosphate-buffered saline (PBS) 1X, pH 7.4 (“PBS”) or (ii) histidine (10 mM), pH 6.0 (“His”). Selected data from this table is presented in Supplementary Figure 10.

DODMA	Initial Time (T0)		4W at 25 °C						4W at 40 °C					
	PBS	His	PBS	PBS	PBS	His	His	His	PBS	PBS	PBS	His	His	His
Degradation	Control	Control	Control	Metals	Peroxides	Control	Metals	Peroxides	Control	Metals	Peroxides	Control	Metals	Peroxides
Oxidation and loss of OC18H36	0	0	0	0	0	0	0.01	0	0	0	0.01	0	0	0
Oxidation and loss of C18H36	0.06	0.06	0.14	0.29	0.28	0.29	0.13	0.28	0.28	0.26	0.12	0.28	0.12	0.28
Oxidation and loss of C19H38	0	0	0	0	0	0	0	0	0	0	0	0	0	0
Loss of C19H38	0.29	0.26	0.54	1.05	1.06	1.08	0.5	1.02	1.05	0.93	0.47	1.04	0.48	1.03
Loss of OC18H34	0	0	0	0	0	0	0	0	0	0	0	0	0	0
Loss of C18H34	0.51	0.28	1.23	2.36	2.29	2.65	1.26	2.63	2.37	2.38	1.32	3.39	1.62	3.39
Double Oxidation (+2O-2H)	0	0	0	0.02	0.01	0	0	0	0.01	0.03	0	0	0	0
Double Oxidation (+2O)	0	0	0	0.03	0.02	0.01	0	0.01	0.02	0.05	0.03	0.01	0.01	0.02
Mono Oxidation (O-2H)	0.04	0.04	0.04	0.32	0.25	0.09	0.04	0.08	0.19	0.37	0.2	0.09	0.05	0.11
Mono Oxidation at Tail 1 (O)	0.05	0.06	0.07	0.12	0.13	0.14	0.08	0.15	0.13	0.15	0.09	0.15	0.1	0.17
Mono Oxidation at Tail 2 (O)	0.05	0.05	0.08	0.16	0.15	0.11	0.06	0.12	0.14	0.18	0.1	0.12	0.07	0.12
Head Group Oxidation	0.51	0.5	0.55	0.47	0.54	0.43	0.44	0.53	0.58	0.53	0.73	0.38	0.37	0.54
Head Group Oxidation and loss of CH ₂	0.21	0.2	0.44	3.48	1.68	0.61	0.36	0.64	2.62	6.25	2.05	0.89	0.64	1.06
Intact	98.26	98.55	96.9	91.69	93.58	94.58	97.12	94.52	92.58	88.85	94.86	93.63	96.54	93.29

Supplementary Table 5. LC/MS analysis of the major DLin-KC2-DMA lipid variant (-28 Da) and degradants after siHPRT-LNP storage over four weeks at 25 or 40 °C in two different buffers: (i) phosphate-buffered saline (PBS) 1X, pH 7.4 (“PBS”) or (ii) histidine (10 mM), pH 6.0 (“His”). Selected data from this table is presented in Supplementary Figure 11.

	Initial Time (T0)		4W at 25 °C						4W at 40 °C					
	PBS	His	PBS	PBS	PBS	His	His	His	PBS	PBS	PBS	His	His	His
Degradations	Control	Control	Control	Metals	Peroxides	Control	Metals	Peroxides	Control	Metals	Peroxides	Control	Metals	Peroxides
Cleavage: C24H36NO	0.02	0.02	0.01	0.17	0.02	0.01	0.02	0.02	0.01	0.22	0.02	0.01	0.02	0.02
Cleavage: C30H55NO4	0	0	0.08	1.47	0.08	0	0.06	0	0.09	2.76	0.13	0	0.16	0
Cleavage: C31H57NO4	0.01	0	0.37	4.51	0.37	0	0.44	0.01	0.35	6.23	0.45	0.01	0.83	0.02
Cleavage: C31H59NO2	0.16	0.18	0.11	0.3	0.1	0.16	0.07	0.17	0.1	0.25	0.11	0.14	0.09	0.14
Cleavage: C32H57NO5	0	0	0.11	2.12	0.11	0	0.94	0	0.06	2	0.08	0	0.6	0
Cleavage: C32H59NO4	0.04	0.02	1.43	10.92	1.41	0.02	2.28	0.03	1.04	11.42	1.3	0.03	2.93	0.05
Cleavage: C33H49NO4	0.01	0.01	0.12	0.67	0.1	0.08	0.09	0.08	0.49	5.74	0.55	0.3	0.38	0.34
Cleavage: C33H59NO3	0.01	0.01	0.25	0.06	0.24	0.01	0.02	0.01	0.29	0.08	0.35	0	0.06	0
Cleavage: C34H61NO3	0.06	0.04	0.38	0.23	0.34	0.03	0.47	0.04	0.16	0.16	0.18	0.03	0.23	0.05
Cleavage: C34H63NO3	0.01	0.01	0.09	0.29	0.08	0.01	0.21	0.02	0.05	0.18	0.06	0.01	0.14	0.02
Cleavage: C35H63NO3	0.01	0.01	0.1	0.24	0.1	0.02	0.05	0.02	0.18	0.08	0.21	0.02	0.31	0.02
Triple Oxidation (3O)	0.06	0.02	0.89	5.73	0.85	0.02	3.63	0.05	0.54	5.02	0.67	0.03	2.48	0.08
Triple Oxidation (3O-2H)	0.05	0.02	1.07	6.74	1.02	0.01	4.07	0.03	0.48	5.91	0.6	0.02	3.06	0.06
Triple Oxidation (3O-4H)	0.02	0	0.62	2.5	0.59	0.01	1.72	0.01	0.27	2.03	0.34	0.01	1.24	0.03
Double Oxidation (2O)	0.73	0.31	5.6	15.83	5.45	0.3	10.18	0.57	3.59	12.47	4.25	0.44	8.17	0.78
Double Oxidation (2O-2H)	0.57	0.27	7.79	12.13	7.54	0.19	9.65	0.28	4.76	10.46	5.49	0.29	8.71	0.51
Double Oxidation (2O-4H)	0.09	0.05	0.85	1.18	0.8	0.07	2.5	0.1	0.48	1.09	0.57	0.12	2.01	0.19
Mono Oxidation (O at Tail 1)	1.65	1.18	6.33	1.09	5.78	1.53	7.83	2.97	4.66	1.19	5.18	2.14	6.48	4.21
Mono Oxidation (O at Tail 2)	0.94	0.81	4.15	7.88	4.08	0.74	6.28	0.88	3.18	5.65	3.3	0.67	6.47	0.85
Head Group Oxidation	0.67	0.3	3.23	5.76	3.31	0.44	6.65	0.91	3.15	8.96	3.68	0.71	8.86	1.62
Oxidation (O-2H)	3.55	2.77	13.79	4.23	13.67	2.6	14.11	3.38	9.78	3.65	10.55	2.41	12.17	3.35
Oxidation and loss of CH2	0.6	0.54	1.76	0.48	1.48	0.58	0.4	0.59	2.17	0.65	2.1	0.76	2.14	0.99
Intact	90.75	93.43	50.85	15.46	52.46	93.17	28.36	89.85	64.12	13.81	59.83	91.85	32.46	86.65

“The Impact of Buffer in RNA-LNP Formulations: Mitigating Oxidation of Ionizable Lipid Tails and RNA-Lipid Adduct Formation”.

Point-by-point responses by the authors (in red) to the reviewers' comments, reproduced verbatim.

Reviewer #4 (Remarks to the Author):

I was asked to provide a 4th review opinion to this manuscript, considering the previous rounds of review and focusing on Reviewer 3's comments.

In short, the authors have addressed prior concerns well, and I recommend to accept the paper for publication.

However, I do have a suggestion. Since the study focuses on siRNA, it would be helpful to mention in the discussion and/or conclusions potential limitations of the study, namely that the findings are most closely related to stability of short, double-stranded siRNA containing LNPs. The data on siRNA should, but may not, completely translate to longer, single-stranded mRNA. Since the introduction and main text heavily notes mRNA LNP vaccine instability as justification, readers will need to understand that the data was accumulated mainly for siRNA. I assume readers will implement this finding on buffers for various mRNA LNPs in the future, so it will be important for them to understand potential limitations as they further investigate these important findings.

We thank the Reviewer for offering a fourth review opinion, including this suggestion. We agree with the Reviewer and have addressed his/her suggestion by including more discussion of the limitations of the study within the main text's Conclusions, namely by supplementing the below section (highlighted new text):

However, a limitation of the work described herein is that it primarily focuses on siRNA-LNPs, which are comparatively more stable than mRNA-LNP systems whose shelf-life is limited by the fragility of the mRNA cargo. While preliminary work demonstrates that the described formulation solutions can be applied more sensitive cargoes, including mRNA, benefits are dependent on ionizable lipid chemistry and are most pronounced with unsaturated lipids (Supplementary Figure 18, Supplementary Table 8). As such, more comprehensive, long-term stability studies using these systems are necessary.

The authors may also consider changing the general “RNA” text in the titles, section headings, and main text to the more specific “siRNA” text since siRNA is the main cargo examined in this paper (other than a small subset of mRNA data in Figure S18 that may be insufficient to establish generalizability).

We have updated the section headings and main text to refer more specifically to “siRNA” or “siRNA-LNPs”, rather than simply “RNA” or “RNA-LNPs”. Additionally, we

have updated the manuscript's title to "Buffer Optimization of siRNA-Lipid Nanoparticles Mitigates Lipid Oxidation and RNA-lipid Adduct Formation" to clarify that the bulk of the findings are focused on siRNA.

This study is highly relevant and well executed. Indeed, the benchmark formulation here is an siRNA LNP that mirrors FDA-approved Onpattro. So the data is good, and I just think readers may need to understand more clearly the work in context of the broad spectrum of RNAs under pre-clinical and clinical development (e.g., siRNA, miRNA, mRNA, tRNA, sgRNA, pegRNA, saRNA, circRNA, etc.), where it seems the major finding relates to data describing siRNA.

Overall, the topic and results are compelling and suitable for publication.